# Towards Generic Semi-Supervised Framework for Volumetric Medical Image Segmentation

**Haonan Wang[1], Xiaomeng Li[1,2]**[*]
[1]The Hong Kong University of Science and Technology
[2]HKUST Shenzhen-Hong Kong Collaborative Innovation Research Institute, Futian, Shenzhen
hwanggr@connect.ust.hk, eexmli@ust.hk

## Abstract

Volume-wise labeling in 3D medical images is a time-consuming task that requires expertise. As a result, there is growing interest in using semi-supervised learning (SSL) techniques to train models with limited labeled data. However, the challenges and practical applications extend beyond SSL to settings such as unsupervised domain adaptation (UDA) and semi-supervised domain generalization (SemiDG). This work aims to develop a generic SSL framework that can handle all three settings. We identify two main obstacles to achieving this goal in the existing SSL framework: 1) the weakness of capturing distribution-invariant features; and 2) the tendency for unlabeled data to be overwhelmed by labeled data, leading to over-fitting to the labeled data during training. To address these issues, we propose an **Aggregating & Decoupling** framework. The aggregating part consists of a Diffusion encoder that constructs a *common knowledge set* by extracting distribution-invariant features from aggregated information from multiple distributions/domains. The decoupling part consists of three decoders that decouple the training process with labeled and unlabeled data, thus avoiding over-fitting to labeled data, specific domains and classes. We evaluate our proposed framework on four benchmark datasets for SSL, Class-imbalanced SSL, UDA and SemiDG. The results showcase notable improvements compared to state-of-the-art methods across all four settings, indicating the potential of our framework to tackle more challenging SSL scenarios. Code and models are available at: https://github.com/xmed-lab/GenericSSL.

## 1   Introduction

Labeling volumetric medical images requires expertise and is a time-consuming process. Therefore, the use of semi-supervised learning (SSL) is highly desirable for training models with limited labeled data. Various SSL techniques [1, 2, 3, 4, 5, 6, 7] have been proposed, particularly in the field of semi-supervised volumetric medical image segmentation (SSVMIS), to leverage both labeled and unlabeled data. However, current SSVMIS methods [8, 9, 10, 11, 12, 13, 14, 15, 16, 17, 18] assume that the labeled and unlabeled data are from the same domain, implying they share the same distribution. In practice, medical images are often collected from different clinical centers using various scanners, resulting in significant domain shifts. These shifts arise due to differences in patient populations, scanners, and scan acquisition settings. As a consequence, these SSVMIS methods have limitations in real-world application scenarios and frequently encounter overfitting issues, leading to suboptimal results.

To address this limitation, researchers have increasingly focused on Unsupervised Domain Adaptation (UDA) techniques. These techniques leverage both labeled (source domain) and unlabeled data (target

---

[*]Corresponding author.

37th Conference on Neural Information Processing Systems (NeurIPS 2023).

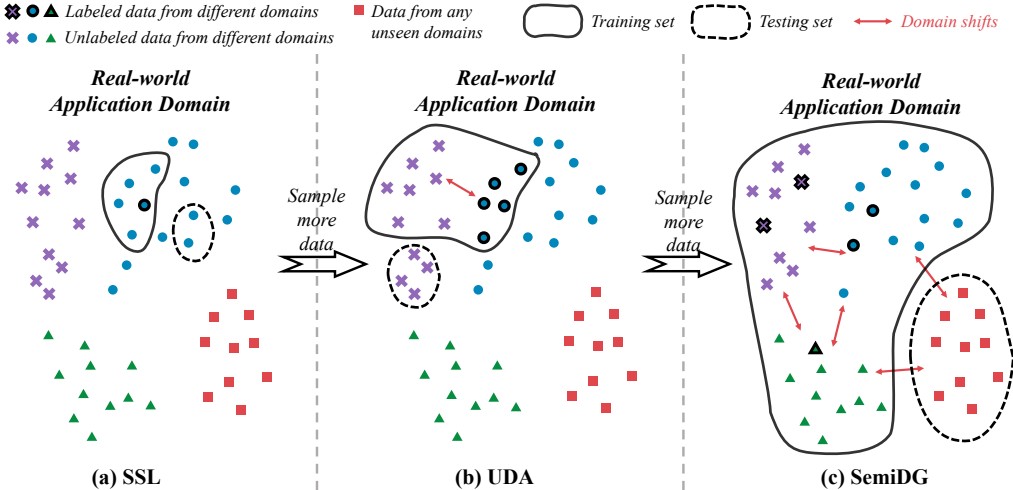

Figure 1: From (a), (b) to (c): generalizing SSL to UDA and SemiDG settings by sampling more diverse data to form the training and testing sets.

domain) for training, but the data originate from different domains. Furthermore, Semi-supervised Domain Generalization (SemiDG), a more stringent scenario, has garnered significant interest. SemiDG utilizes labeled and unlabeled data from multiple domains during training and is evaluated on an unseen domain. Currently, methods for these three scenarios are optimized separately, and there is no existing approach that addresses all three scenarios within a unified framework. However, given that all training stages involve labeled and unlabeled data, it is intuitive to explore a **generic SSL-based framework** that can handle all settings and eliminate the need for complex task-specific designs. Therefore, this paper aims to develop a generic framework that can handle existing challenges in real-world scenarios, including:

- *Scenario 1: SSL (Figure 1(a)): The sample data used for both training and testing are from the same domain, representing the standard SSL setting.*
- *Scenario 2: UDA (Figure 1(b)): The sampled data originate from two domains, with the labels of the target domain being inaccessible, representing the UDA setting.*
- *Scenario 3: SemiDG (Figure 1(c)): The sampled data encompasses multiple domains, with only a limited number of them being labeled, representing the SemiDG setting.*

Potential similarities can be found and summarized as follows: (1) in the training stage, both labeled data and unlabeled data are used; (2) in the scenario of the real-world application domain, whether the distribution shifts in SSL or the domain shifts in UDA and SemiDG can all be regarded as *sampling bias*, *i.e.*, the main difference is how we sample the data in Figure 1.

Now we wonder whether the existing SSVMIS methods are powerful enough to handle this general task. Experimental results show that the existing SSL methods do not work well on UDA and SemiDG settings, as shown in Table 3 & 4, and vice versa (Table 2). *One of the main obstacles lies in the severe over-fitting of these models, which is caused by the dominance of labeled data during training*. Specifically, the state-of-the-art SSVMIS methods are mainly based on two frameworks: (1) Teacher-student framework [1], where a student model is first trained with the labeled data, and a teacher model obtained from the EMA of the student model generates the pseudo label to re-train the student model with labeled data, see Figure 2(a); (2) CPS (Cross Pseudo Supervision) [3] framework, which leverages the consistency between two perturbed models and the pseudo label generated by one of the networks will be used to train the other network, see Figure 2(b). The predicting modules in these two main frameworks are trained with both labeled and unlabeled data; however, the labeled data, with precise ground truths as supervision, converges more rapidly compared with the unlabeled data. **Thus, the training process will easily get overwhelmed by the supervised training task**, as shown in Figure 3. Another challenge lies in the fact that existing SSVMIS methods fail to address the issue of distribution shifts, let alone domain shifts, resulting in a limitation in capturing features that are invariant to changes in distribution.

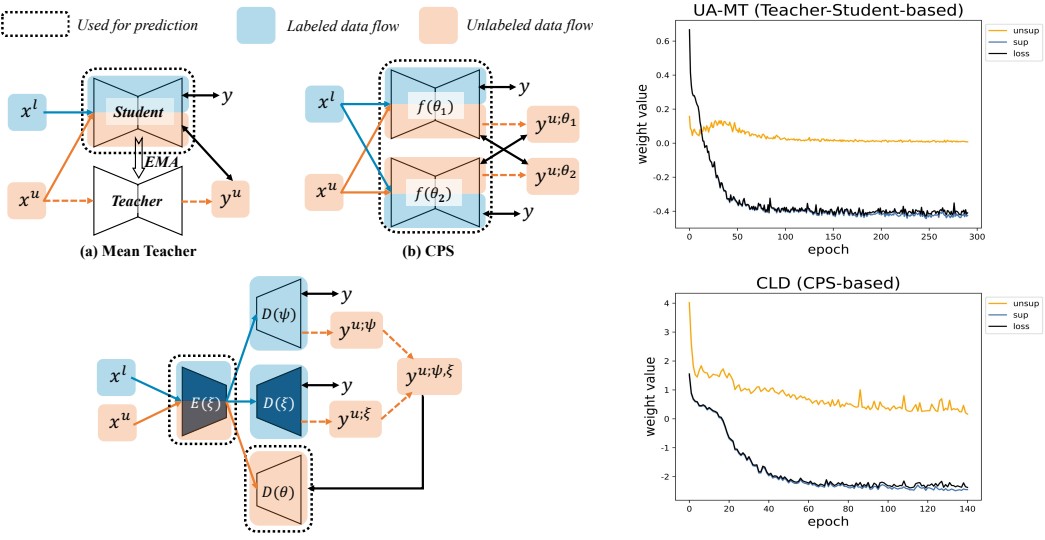

Figure 2: Our proposed A&D framework differs in that the labeled and unlabeled data flows are separate, and only the unlabeled flow is used for predicting.

Figure 3: The training loss curves of UA-MT and CLD on MMWHS dataset. The overall losses (black) are dominated by the supervised losses.

Based on the similarities and the main weaknesses of the mainstream SSVMIS methods, we argue that a generic framework is possible if we can solve the over-fitting issue and design a powerful methods to capture the distribution-invariant features. To tackle the above issues and design a generic SSVMIS methods for the real-world application scenarios, this work proposes a novel Aggregating & Decoupling (A&D) framework. Specifically, A&D consists of an Aggregating stage and a Decoupling stage. In the Aggregating stage, based on the recent success of the diffusion model [19, 20], we propose a Diff-VNet to aggregate the multi-domain features into one shared encoder to construct a *common knowledge set* to improve the capacity of capturing the distribution-invariant features. To solve the over-fitting issue, in the Decoupling stage, we decouple the decoding process to (1) a labeled data training flow which mainly updates a Diff-VNet decoder and a difficulty-aware V-Net decoder to generate high-quality pseudo labels; (2) an unlabeled data training flow which mainly updates another vanilla V-Net decoder with the supervision of the pseudo labels. The denoising process of the Diff-VNet decoder provides the domain-unbiased pseudo labels while the difficulty-aware V-Net decoder class-unbiased pseudo labels. We also propose a re-parameterizing & smoothing combination strategy to further improve the quality of the pseudo labels.

The key contributions of our work can be summarized as follows: (1) we unify the SSL, Class Imbalanced SSL, UDA, and SemiDG for volumetric medical image segmentation with one generic framework; (2) we state the over-fitting issues of the current SSL methods and propose to solve it by an efficient data augmentation strategy and decoupling the decoders for labeled data and unlabeled, respectively; (3) we introduce the Diffusion V-Net to learn the underlying feature distribution from different domains to generalize the SSL methods to more realistic application scenarios; (4) The proposed Aggregating & Decoupling framework achieves state-of-the-art on representative datasets of SSL, class-imbalance SSL, UDA, and SemiDG tasks. Notably, our method achieves significant improvements on the Synapse dataset (12.3 in Dice) and the MMWHS dataset in the MR to CT setting (8.5 in Dice). Extensive ablation studies are conducted to validate the effectiveness of the proposed methods.

## 2 Related Work

### 2.1 Semi-supervised Segmentation & the Class Imbalance Issue

Semi-supervised segmentation aims to explore tremendous unlabeled data with supervision from limited labeled data. Recently, self-training-based methods [3, 4, 21] have become the mainstream

of this domain. Approaches with consistency regularization strategies [22, 3, 21] achieved good performance by encouraging high similarity between the predictions of two perturbed networks for the same input image, which highly improved the generalization ability. In the medical image domain, the data limitation issue is more natural and serious. Existing approaches [23, 24, 10, 14, 13, 17, 16, 25] to combat the limited data have achieved great success but are bottlenecked by the application scenarios and cannot handle more challenging but practical settings such as UDA and SemiDG.

**Class Imbalance Issue**   Class imbalance issue is a significant problem to extend the existing SSL-based methods to more practical setting, since medical datasets have some classes with notably higher instances in training samples than others. In natural image domain, different means are proposed to solve this issue, including leveraging unlabeled data [26, 27, 28, 29, 30], re-balancing data distributions in loss [31, 30, 32], debiased learning [6, 5] *etc.* In medical image domain, this issue is more severe but only few work [15, 33, 25] noticed this problem. Incorporating the class-imbalance awareness is crucial for the generalization of the SSL methods.

## 2.2   Unsupervised Domain Adaptation & Semi-supervised Domain Generalization

Domain adaptation (DA) aims to solve the domain shifts by jointly training the model with source domain data and target domain data. Unsupervised Domain Adaptation (UDA) [34, 35, 36, 37] is the most challenging and practical setting among all the DA setting, since no target labels are required. In this context, UDA is becoming increasingly important in the medical image segmentation field, and as a result, a myriad of UDA approaches have been developed for cross-domain medical image segmentation by leveraging: generative adversarial-based methods [38, 39, 40, 41, 42, 43, 44, 45, 46], semi-supervised learning techniques [47, 48, 49], and self-training as well as contrastive learning techniques [50, 51], *etc.* Though with promising adaptation results, these methods highly rely on the unlabeled target domain information, which hinders the generalizability.

Domain generalization (DG) is a more strict setting, where the difference with DA is that the model does not use any information from the target domain. Unlike unsupervised domain adaptation, semi-supervised domain generalization (SemiDG) does not assume access to labeled data from the target domains. Existing SemiDG methods [52, 53] leverage various unusual strategies to solve the domain shifts, *e.g.*, meta-learning [52], Fourier Transformation [54], compositionality [55] *etc.*, which are not general and have unsatisfactory performance on the tasks such as UDA and SSL.

Compared to these prior works, our work is the first to unify SSL, Imbalanced SSL, UDA, and SemiDG settings. This extension not only amplifies the scope and versatility of SSL-based frameworks in medical image segmentation but also stands in stark contrast to earlier approaches that remained confined to singular domains such as SSL or UDA.

## 2.3   Diffusion Model

Denoising diffusion models [56, 19, 57, 58] have shown significant success in various generative tasks [59, 60, 61, 62, 63], due to the powerful ability of modeling the underlying distribution of the data, conceptually having a greater capacity to handle challenging tasks. Noticing this property, there has been a rise in interest to incorporate them into segmentation tasks, including both natural image segmentation [64, 65, 66], and medical image segmentation [67, 68, 20] Given the notable achievements of diffusion models in these respective domains, leveraging such models to develop generation-based perceptual models would prove to be a highly promising avenue to push the boundaries of perceptual tasks to newer heights.

## 3   Method

### 3.1   Overview of the Aggregating & Decoupling Framework

In this section, we will introduce our **Aggregating & Decoupling (A&D)** framework, as shown in Figure 4, which consists of an Aggregating stage and a Decoupling stage. The training pipeline is illustrated in Algorithm 1.

The **Aggregating stage** aims to construct a *common knowledge set* across domains based on the idea that all data share common underlying high-level knowledge, such as texture information.

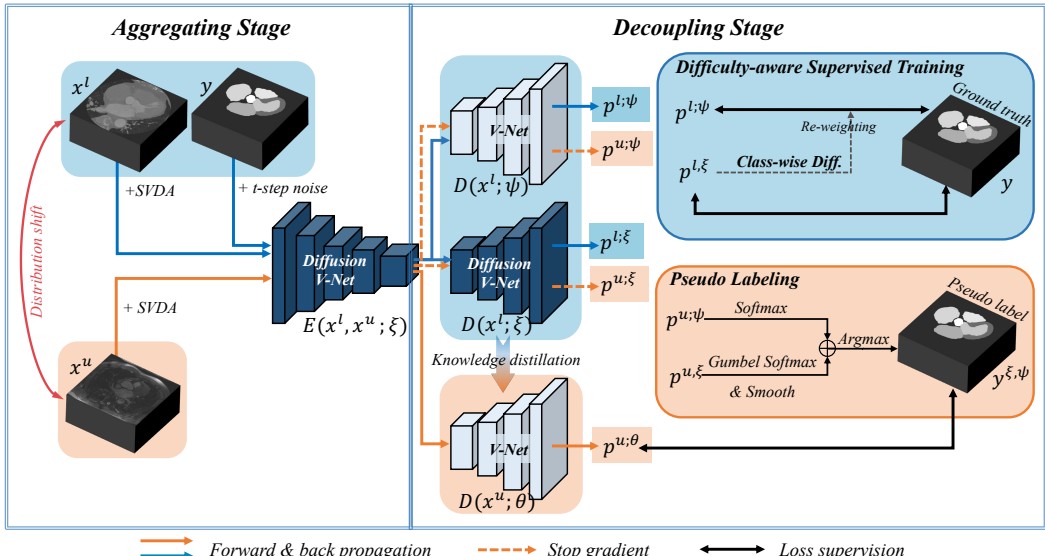

Figure 4: Overview of the proposed **Aggregating & Decoupling** framework. Blue and orange regions denote the training process with labeled data and unlabeled data, respectively. We separate the training of the decoders using labeled data and unlabeled data, and only use the decoder trained with unlabeled data for prediction.

By aggregating information from multiple domains and jointly trained, the encoder can capture the underlying domain-invariant features. To achieve this, we introduce a powerful yet efficient Sampling-based Volumetric Data Augmentation (SVDA) strategy to enlarge the distribution diversity and leverage the diffusion model to capture the invariant features of the diversified data.

In the decoding of the existing SSL methods, the decoders are simultaneously trained with both labeled and unlabeled data, which leads to coupling and over-fitting issues and further hinder the extending to the general SSL. The **Decoupling stage** aims to solve these issues by decoupling the labeled and unlabeled data training flows. Concretely, for the labeled data flow, (1) a diffusion decoder is mainly used to guide the diffusion encoder to learn the distribution-invariant representations through the diffusion backward process, and thus produces the *domain-unbiased pseudo labels*; (2) a vanilla V-Net decoder with the proposed difficulty-aware re-weighting strategy is mainly to avoid the model over-fit to the easy and majority classes, and thus produces the *class-unbiased pseudo labels*. Then, for the unlabeled data flow, the domain- and class-unbiased pseudo labels are ensembled through a proposed Reparameterize & Smooth (RS) strategy to generate high quality pseudo labels. Finally, the pseudo labels are used to supervise the training of an additional V-Net decoder for prediction only.

### 3.2 Aggregating Stage

Assume that the entire dataset comprises of $N_L$ labeled samples $\{(x_i^l, y_i)\}_{i=1}^{N_L}$ and $N_U$ unlabeled samples $\{x_i^u\}_{i=1}^{N_U}$, where $x_i \in \mathbb{R}^{D \times H \times W}$ is the input volume and $y_i \in \mathbb{R}^{K \times D \times H \times W}$ is the ground-truth annotation with $K$ classes. The goal of the aggregating stage is to augment the data with SVDA and encode the labeled $(x^l, y)$ and unlabeled data $x^u$ to high-level distribution-invariant features for denoising labeled data flow $h^{l;\xi}$ difficulty-aware labeled data flow $h^{l;\psi}$, and unlabeled data flow $h^u$.

**Sampling-based Volumetric Data Augmentation (SVDA)**  Instead of the time-consuming traditional data augmentation used in [69] which cascaded all the augmentations, SVDA build upon an augmentation set and $N_{aug}$ operations are randomly sampled to apply to both the labeled and unlabeled data. The augmentation set consists of 3D spatial-wise transforms (random crop, random rotation, random scaling) and voxel-wise transforms (Gaussian blur, brightness, contrast, gamma). $N_{aug}$ is empirically set to 3.

---

**Algorithm 1:** Training Pipeline of A&D.

---

**Input:** Labeled samples $\{(x_i^l, y_i)\}_{i=1}^{N_L}$, unlabeled samples $\{x_i^u\}_{i=1}^{N_U}$
**Output:** Diffusion encoder $E(x^l, x^u; \xi)$ with V-Net decoder $D(x^u; \theta)$ for inference

---

1   Initialization;
2   **for** *batched data* $(x_i^l, y_i)$, $x_i^u$ **do**
3      Add SVDA on $x^l$ and $x^u$ to obtain $\hat{x}^l$ and $\hat{x}^u$;
4      Add $t$ step noise on $y_i$ to obtain the noisy label $y_t$ by Eq. 1;
5      Train the diffusion model $E(x^l; \xi)$ and $D(x^l; \xi)$ with $(\hat{x}_i^l, y_t)$ by Eq. 2;
6      Train the diffusion encoder $E(x^l; \xi)$ and difficulty-aware decoder $D(x^l; \psi)$ by Eq. 6;
7      Generate *domain-unbiased* probability map $p^{u;\xi}$ with $E(x^l, x^u; \xi)$+$D(x^l; \xi)$ by DDIM [57];
8      Generate *class-unbiased* probability map $p^{u;\psi}$ with $D(x^l; \psi)$ with forward pass;
9      Ensemble the two maps and obtain the pseudo label $y^{\xi,\psi}$ by Eq. 7;
10     Train $E(x^u; \xi)$ and V-Net decoder $D(x^u; \theta)$ with unlabeled data $(\hat{x}^u, y^{\xi,\psi})$ and by Eq. 8;
11 **end**

---

**Diffusion for Capturing Invariant Features**   We follow Diff-UNet [20] to use diffusion model for perception but modify it to a V-Net version and remove the additional image encoder. Given the labeled volume data $x^l \in \mathbb{R}^{D \times W \times H}$ and its label $y \in \mathbb{R}^{D \times W \times H}$, we first convert the label to the one-hot format $y_0 \in \mathbb{R}^{K \times D \times W \times H}$ and add successive $t$ step noise $\epsilon$ to obtain the noisy label $y_t \in \mathbb{R}^{K \times D \times W \times H}$, which is the diffusion forward process:

$$y_t = \sqrt{\bar{\alpha}_t} y_0 + \sqrt{1 - \bar{\alpha}_t} \epsilon, \epsilon \in \mathcal{N}(0, 1) \tag{1}$$

Then, the noisy label is concatenated with the image $x^l$ as the input of the Diff-VNet. Concretely, the high-level features are different for different data flows. For the denoising flow, *i.e.*, $D(x^l; \xi)$ as decoder, the diffusion encoder takes concatenation $\texttt{concat}([y_t, x^l])$ and time step $t$ as input to generate the time-step-embedded multi-scale features $h_i^{l;\xi} \in \mathbb{R}^{i \times F \times \frac{D}{2^i} \times \frac{W}{2^i} \times \frac{H}{2^i}}$ where $i$ is the stage and $F$ is the basic feature size. $h_i^{l;\xi}$ are further used by $D(x^l; \xi)$ to predict the clear label $y_0$. For the difficulty-aware training flow and the unlabeled data flow, *i.e.*, $D(x^l; \psi)$ and $D(x^u; \theta)$ as decoders, the encoder only takes $x_l$ and $x_u$ as input to obtain the multi-scale features $h_i^{l;\psi}$, $h_i^u$, respectively. Note that $h_i^{l;\psi}$, $h_i^u$ are with same shapes with $h_i^{l;\xi}$.

### 3.3   Decoupling Stage

The decoupling stage consists of four steps: supervised denoising training to generate domain-unbiased pseudo labels, supervised difficulty-aware training to generate class-unbiased pseudo labels, pseudo labeling to ensemble the two pseudo labels and unsupervised training to get the final predictor.

**Supervised Denoising Training with the Diffusion Decoder** $D(x^l; \xi)$   Taking $h_i^{l;\xi}$ as inputs, $D(x^l; \xi)$ decodes the features to predict the clear label $y_0$ as domain-unbiased pseudo label. The objective function is defined as follow:

$$\mathcal{L}_{deno} = \frac{1}{N_L} \sum_{i=0}^{N_L} \mathcal{L}_{DiceCE}(p_i^{l;\xi}, y_i) \tag{2}$$

where $\mathcal{L}_{DiceCE}(x, y) = \frac{1}{2}[\mathcal{L}_{CE}(x, y) + \mathcal{L}_{Dice}(x, y)]$ is the combined dice and cross entropy loss.

**Supervised Difficulty-aware Training with** $D(x^l; \psi)$   To alleviate the common class imbalance issue in SSVMIS, we design a Difficulty-aware Re-weighting Strategy (DRS) to force the model to focus on the most difficult classes (*i.e.* the classes learned slower and with worse performances). The difficulty is modeled in two ways with the probability map $p^{l;\xi}$ produced by diffusion decoder $D(x^l; \xi)$: learning speed and performance. We use Population Stability Index [70] to measure the

learning speed of each class after the $e^{th}$ iteration:

$$du_{k,e} = \sum_{e-\tau}^{e} \min(\triangle, 0)\ln(\frac{\lambda_{k,e}}{\lambda_{k,e-1}}), \quad dl_{k,e} = \sum_{e-\tau}^{e} \max(\triangle, 0)\ln(\frac{\lambda_{k,e}}{\lambda_{k,e-1}}) \quad (3)$$

where $\lambda_k$ denotes the Dice score of $p^{l;\xi}$ of $k^{th}$ class in $e^{th}$ iteration and $\triangle = \lambda_{k,e} - \lambda_{k,e-1}$. $du_{k,e}$ and $dl_{k,e}$ denote classes not learned and learned after $e^{th}$ iteration. $\tau$ is the number accumulation iterations and set to 50 empirically. Then, we define the difficulty of $k^{th}$ class after $e^{th}$ iteration as:

$$d_{k,e} = \frac{du_{k,e}}{dl_{k,e}} \quad (4)$$

where the classes learned faster have smaller $d_{k,e}$, the corresponding weights in the loss function will be smaller to slow down the learn speed. We also accumulate $1 - \lambda_{k,e}$ for $\tau$ iterations to obtain the reversed dice weight $w_{\lambda_{k,e}}$ and weight $d_{k,e}$. In this case, classes with lower dice scores will have larger weights in the loss function, which forces the model to pay more attention to these classes. The overall difficulty-aware weight of $k^{th}$ class is defined as:

$$w_k^{diff} = w_{\lambda_{k,e}} \cdot (d_{k,e})^{\alpha} \quad (5)$$

where $\alpha$ is empirically set to $\frac{1}{5}$ in the experiments to alleviate outliers. The objective function of the supervised difficulty-aware training is defined as follow:

$$\mathcal{L}_{diff} = \frac{1}{N_L} \frac{1}{K} \sum_{i=0}^{N_L} \sum_{k=0}^{K} w_k^{diff} \mathcal{L}_{DiceCE}(p_{i,k}^{l;\psi}, y_{i,k}) \quad (6)$$

**Pseudo Labeling with Reparameterize & Smooth (RS) Strategy**    The domain-unbiased $p^{u;\xi}$ probability map is generated by iterating the diffusion model ($E(x^l, x^u; \xi) + D(x^l; \xi)$) $t$ times with the Denoising Diffusion Implicit Models (DDIM) method [57]. The class-unbiased $p^{u;\psi}$ probability map can be obtained by $D(x^l; \psi)$ with stopped gradient forward pass. We ensemble $p^{u;\xi}$ and $p^{u;\psi}$ to generate high-quality pseudo labels. However, when combining these two maps, we found that the denoised probability map $p^{u;\psi}$ is too sparse, *i.e.*, with very high confidence of each class. This property is benefit for the fully-supervised tasks, but in this situation, it will suppress $p^{u;\psi}$ and is not robust to noise and error. Thus, we re-parameterize $p^{u;\psi}$ with the Gumbel-Softmax to add some randomness and using Gaussian blur kernel to remove the noise brought by this operation. The final pseudo label is:

$$y^{\xi,\psi} = \texttt{argmax}(\texttt{Gumbel-Softmax}(p^{u;\xi}) + \texttt{Softmax}(p^{u;\psi})) \quad (7)$$

**Unsupervised Training with $D(x^u; \theta)$**    Finally, we can use the pseudo label $y^{\xi,\psi}$ to train $D(x^u; \theta)$ in an unsupervised manner. The objective function of the unsupervised training is defined as:

$$\mathcal{L}_u = \frac{1}{N_U} \sum_{i=0}^{N_U} \mathcal{L}_{DiceCE}(p_i^{u;\theta}, y^{\xi,\psi}) \quad (8)$$

To better leverage the domain- and class-unbiased features, we also transmit with knowledge distillation strategy: $\theta = w_{ema} \times \theta + (1 - w_{ema}) \times (\xi + \psi)/2, w_{ema} = 0.99$. The overall training function of the A&G framework is:

$$\mathcal{L} = \mathcal{L}_{deno} + \mathcal{L}_{diff} + \mu \mathcal{L}_u \quad (9)$$

where $\mu$ is empirically set as 10 and follow [15] to use the epoch-dependent Gaussian ramp-up strategy to gradually enlarge the ratio of unsupervised loss. In the inference stage, only the diffusion encoder $E(x^l, x^u; \xi)$ and $D(x^u; \theta)$ are used to generate the predictions.

## 4    Experiments

### 4.1    Datasets and Implementation Details

We evaluate our proposed A&D framework on four datasets for four tasks, *i.e.*, LASeg dataset [72] for SSL, Synapse dataset [73] for class imbalanced SSL, MMWHS dataset [74] for UDA, and M&Ms

Table 1: Results on Synapse dataset with 20% labeled data for **class imbalanced SSL** task. 'Common' or 'Imbalance' indicates whether the methods consider the imbalance issue or not. Notably, our method does not introduce many additional parameters compared with the existing SSL methods. Results of 3-times repeated experiments are reported in 'mean±std' format. Best results are boldfaced.

| | Methods | Avg. Dice | Avg. ASD | Sp | RK | LK | Ga | Es | Li | St | Ao | IVC | PSV | PA | RAG | LAG |
|---|---|---|---|---|---|---|---|---|---|---|---|---|---|---|---|---|
| | V-Net (fully) | 62.09±1.2 | 10.28±3.9 | 84.6 | 77.2 | 73.8 | 73.3 | 38.2 | 94.6 | 68.4 | 72.1 | 71.2 | 58.2 | 48.5 | 17.9 | 29.0 |
| General | UA-MT [8] | 20.26±2.2 | 71.67±7.4 | 48.2 | 31.7 | 22.2 | 0.0 | 0.0 | 81.2 | 29.1 | 23.3 | 27.5 | 0.0 | 0.0 | 0.0 | 0.0 |
| General | URPC [10] | 25.68±5.1 | 72.74±15.5 | 66.7 | 38.2 | 56.8 | 0.0 | 0.0 | 85.3 | 33.9 | 33.1 | 14.8 | 0.0 | 5.1 | 0.0 | 0.0 |
| General | CPS [3] | 33.55±3.7 | 41.21±9.1 | 62.8 | 55.2 | 45.4 | 35.9 | 0.0 | 91.1 | 31.3 | 41.9 | 49.2 | 8.8 | 14.5 | 0.0 | 0.0 |
| General | SS-Net [17] | 35.08±2.8 | 50.81±6.5 | 62.7 | 67.9 | 60.9 | 34.3 | 0.0 | 89.9 | 20.9 | 61.7 | 44.8 | 0.0 | 8.7 | 4.2 | 0.0 |
| General | DST [6] | 34.47±1.6 | 37.69±2.9 | 57.7 | 57.2 | 46.4 | 43.7 | 0.0 | 89.0 | 33.9 | 43.3 | 46.9 | 9.0 | 21.0 | 0.0 | 0.0 |
| General | DePL [5] | 36.27±0.9 | 36.02±0.8 | 62.8 | 61.0 | 48.2 | 54.8 | 0.0 | 90.2 | 36.0 | 42.5 | 48.2 | 10.7 | 17.0 | 0.0 | 0.0 |
| Imbalance | Adsh [71] | 35.29±0.5 | 39.61±4.6 | 55.1 | 59.6 | 45.8 | 52.2 | 0.0 | 89.4 | 32.8 | 47.6 | 53.0 | 8.9 | 14.4 | 0.0 | 0.0 |
| Imbalance | CReST [26] | 38.33±3.4 | 22.85±9.0 | 62.1 | 64.7 | 53.8 | 43.8 | 8.1 | 85.9 | 27.2 | 54.4 | 47.7 | 14.4 | 13.0 | 18.7 | 4.6 |
| Imbalance | SimiS [28] | 40.07±0.6 | 32.98±0.5 | 62.3 | 69.4 | 50.7 | 61.4 | 0.0 | 87.0 | 33.0 | 59.0 | 57.2 | 29.2 | 11.8 | 0.0 | 0.0 |
| Imbalance | Basak et al. [33] | 33.24±0.6 | 43.78±2.5 | 57.4 | 53.8 | 48.5 | 46.9 | 0.0 | 87.8 | 28.7 | 42.3 | 45.4 | 6.3 | 15.0 | 0.0 | 0.0 |
| Imbalance | CLD [15] | 41.07±1.2 | 32.15±3.3 | 62.0 | 66.0 | 59.3 | 61.5 | 0.0 | 89.0 | 31.7 | 62.8 | 49.4 | 28.6 | 18.5 | 0.0 | 5.0 |
| Imbalance | DHC [25] | 48.61±0.9 | 10.71±2.6 | 62.8 | 69.5 | 59.2 | **66.0** | 13.2 | 85.2 | 36.9 | 67.9 | 61.5 | 37.0 | 30.9 | 31.4 | 10.6 |
| Imbalance | **A&D (ours)** | **60.88±0.7** | **2.52±0.4** | **85.2** | 66.9 | 67.0 | 52.7 | **62.9** | 89.6 | 52.1 | 83.0 | 74.9 | 41.8 | 43.4 | 44.8 | 27.2 |

Table 2: Results on two settings of LASeg dataset for **SSL** task.

| 5% labeled data (labeled:unlabeled=4:76) | | | | |
|---|---|---|---|---|
| Method | | Metrics | | |
| | Dice | Jaccard | 95HD | ASD |
| V-Net (fully) | 91.47 | 84.36 | 5.48 | 1.51 |
| V-Net (5%) | 52.55 | 39.60 | 47.05 | 9.87 |
| UA-MT [8] | 82.26 | 70.98 | 13.71 | 3.82 |
| SASSNet [9] | 81.60 | 69.63 | 16.16 | 3.58 |
| DTC [11] | 81.25 | 69.33 | 14.90 | 3.99 |
| URPC [10] | 82.48 | 71.35 | 14.65 | 3.65 |
| MC-Net [12] | 83.59 | 72.36 | 14.07 | 2.70 |
| SS-Net [17]† | 86.33 | 76.15 | 9.97 | 2.31 |
| BCP [76]† | 88.02 | 78.72 | 7.90 | 2.15 |
| **A&D (ours)** | **89.93** | **81.82** | **5.25** | **1.86** |
| 10% labeled data (labeled:unlabeled=8:72) | | | | |
| Method | | Metrics | | |
| | Dice | Jaccard | 95HD | ASD |
| V-Net (10%) | 82.74 | 71.72 | 13.35 | 3.26 |
| UA-MT [8] | 87.79 | 78.39 | 8.68 | 2.12 |
| SASSNet [9] | 87.54 | 78.05 | 9.84 | 2.59 |
| DTC [11] | 87.51 | 78.17 | 8.23 | 2.36 |
| URPC [10] | 86.92 | 77.03 | 11.13 | 2.28 |
| LMISA-3D [77]* | 86.06 | 76.53 | 12.99 | 2.41 |
| vMFNet [55]* | 73.88 | 62.56 | 16.81 | 5.04 |
| MC-Net [12] | 87.62 | 78.25 | 10.03 | 1.82 |
| SS-Net [17]† | 88.55 | 79.62 | 7.49 | 1.90 |
| BCP [76]† | 89.62 | 81.31 | 6.81 | 1.76 |
| **A&D (ours)** | **90.31** | **82.40** | **5.55** | **1.64** |

† use test set for validation, we use labeled data instead.
★ SOTA methods tailored for UDA and SemiDG, respectively.

Table 3: Results on two settings of MMWHS dataset for **UDA** task.

| MR to CT | | | | | | |
|---|---|---|---|---|---|---|
| Method | | | Dice | | | ASD |
| | AA | LAC | LVC | MYO | Average | Average |
| Supervised Training | 92.7 | 91.1 | 91.9 | 87.8 | 90.9 | 2.2 |
| PnP-AdaNet [38] | 74.0 | 68.9 | 61.9 | 50.8 | 63.9 | 12.8 |
| AdaOutput [37] | 65.2 | 76.6 | 54.4 | 43.6 | 59.9 | 9.6 |
| CycleGAN [34] | 73.8 | 75.7 | 52.3 | 28.7 | 57.6 | 10.8 |
| CyCADA [36] | 72.9 | 77.0 | 62.4 | 45.3 | 64.4 | 9.4 |
| SIFA [41] | 81.3 | 79.5 | 73.8 | 61.6 | 74.1 | 7.0 |
| DSFN [42] | 84.7 | 76.9 | 79.1 | 62.4 | 75.8 | N/A |
| DSAN [43] | 79.9 | 84.8 | 82.8 | 66.5 | 78.5 | 5.9 |
| LMISA-3D [77] | 84.5 | 82.8 | 88.6 | 70.1 | 81.5 | 2.3 |
| **A&D (ours)** | **93.2** | **89.5** | **91.7** | **86.2** | **90.1** | **1.7** |
| CT to MR | | | | | | |
| Method | | | Dice | | | ASD |
| | AA | LAC | LVC | MYO | Average | Average |
| Supervised Training | 82.8 | 80.5 | 92.4 | 78.8 | 83.6 | 2.9 |
| PnP-AdaNet [38] | 43.7 | 68.9 | 61.9 | 50.8 | 63.9 | 8.9 |
| AdaOutput [37] | 60.8 | 39.8 | 71.5 | 35.5 | 51.9 | 5.7 |
| CycleGAN [34] | 64.3 | 30.7 | 65.0 | 43.0 | 50.7 | 6.6 |
| CyCADA [36] | 60.5 | 44.0 | 77.6 | 47.9 | 57.5 | 7.9 |
| SIFA [41] | 65.3 | 62.3 | 78.9 | 47.3 | 63.4 | 5.7 |
| DSAN [43] | **71.3** | 66.2 | 76.2 | 52.1 | 66.5 | 5.4 |
| LMISA-3D [77] | 60.7 | 72.4 | **86.2** | 64.1 | 70.8 | **3.6** |
| SS-Net [17]* | 62.1 | 58.4 | 68.9 | 51.4 | 60.2 | 5.9 |
| BCP [76]* | 63.6 | 63.7 | 70.9 | 58.0 | 64.1 | 4.5 |
| **A&D (ours)** | 62.8 | **87.4** | 61.3 | **74.1** | **71.4** | 7.9 |

★ SOTA methods on semi-supervised segmentation.

dataset [75] for SemiDG. For more details, please refer to the Appendix. We implement the proposed framework with PyTorch, using a single NVIDIA A100 GPU. The network parameters are optimized by SGD with Nesterov and momentum of 0.9. We employ a "poly" decay strategy follow [69]. For more implementation details, *e.g.*, data preprocessing, learning rates, batch sizes, *etc.*, please refer to the Appendix. We evaluate the prediction of the network with two metrics, including Dice and the average surface distance (ASD). For the LASeg dataset, we employ additional metrics, Jaccard and HD95, following [8]. Final segmentation results are obtained using a sliding window strategy.

## 4.2 Experiment results

Our proposed A&D framework achieves state-of-the-art on all the four settings, *i.e.*, SSL (Table 2), imbalanced SSL (Table 1), UDA (Table 3), and SemiDG (Table 4). In particular, our method achieves

Table 4: Results on two settings of M&Ms dataset for **SemiDG** task.

| Method | 2% Labeled data | | | | | 5% Labeled data | | | | |
|---|---|---|---|---|---|---|---|---|---|---|
| | Domain A | Domain B | Domain C | Domain D | Average | Domain A | Domain B | Domain C | Domain D | Average |
| nnUNet [69] | 52.87 | 64.63 | 72.97 | 73.27 | 65.94 | 65.30 | 79.73 | 78.06 | 81.25 | 76.09 |
| SDNet+Aug [53] | 54.48 | 67.81 | 76.46 | 74.35 | 68.28 | 71.21 | 77.31 | 81.40 | 79.95 | 77.47 |
| LDDG [78] | 59.47 | 56.16 | 68.21 | 68.56 | 63.16 | 66.22 | 69.49 | 73.40 | 75.66 | 71.29 |
| SAML [79] | 56.31 | 56.32 | 75.70 | 69.94 | 64.57 | 67.11 | 76.35 | 77.43 | 78.64 | 74.88 |
| BCP [76]* | 71.57 | 76.20 | 76.87 | 77.94 | 75.65 | 73.66 | 79.04 | 77.01 | 78.49 | 77.05 |
| DGNet [52] | 66.01 | 72.72 | 77.54 | 75.14 | 72.85 | 72.40 | 80.30 | 82.51 | 83.77 | 79.75 |
| vMFNet [55] | 73.13 | 77.01 | **81.57** | 82.02 | 78.43 | 77.06 | 82.29 | **84.01** | **85.13** | 82.12 |
| **A&D (ours)** | **79.62** | **82.26** | 80.03 | **83.31** | **81.31** | **81.71** | **85.44** | 82.18 | 83.9 | **83.31** |

* SOTA method on semi-supervised segmentation.

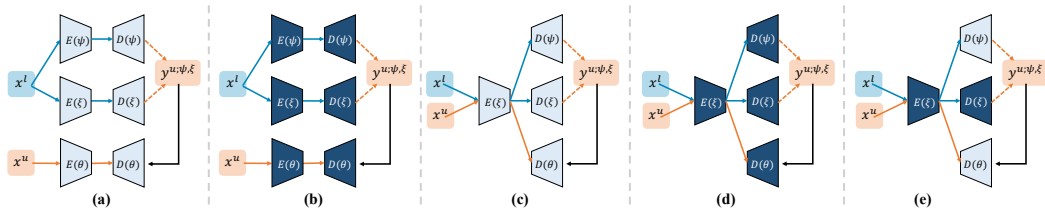

Figure 5: Ablation study on different architectures. (e) is the final framework

significant improvements over the existing state-of-the-art approach on the Synapse dataset with 20% labeled data and the MMWHS dataset in the MR to CT setting, demonstrating a substantial increase of 12.3 in Dice score for the Synapse dataset and 8.5 for the MMWHS dataset.

## 4.3 Analyses

**Architecture Analysis** As shown in Figure 5, we test different architectures for the framework, the corresponding results are in Table 5. When using three separate networks (a)(b), the performance drop significantly, the reason is that this structure can not learn the domain-invariant features. Using pure V-Net (c) or diffusion-based networks (d) also leads to inferior results, especially for the pure diffusion model, it is hard to train due to the limited labeled data, high-quality pseudo labels are hard to obtain, which further hinder the training of the unsupervised branch.

Table 5: Ablation study on the MMWHS MR to CT setting of different architectures with respect to those in Figure 5.

| Arch. | a | b | c | d | e |
|---|---|---|---|---|---|
| Dice | 76.14 | 50.53 | 82.19 | 45.63 | 90.14 |

**Ablation on the components** We analyze the effectiveness of different components in our method. According to the results in Table 6, on MMWHS MR to CT setting (UDA), when removing the diffusion model, performance decreases the most, which indicated the importance ability of the diffusion model for capturing the distribution-invariant features. The result of removing the SVDA also indicates that when the data is not diverse enough, the diffusion model cannot capture effective underlying features. On Synapse dataset for IBSSL, the results are slightly different with those in the UDA setting, the DRS plays more important role than the Diffusion. In 5% M&Ms dataset for SemiDG, the results are quite aligned with results in the UDA setting.

Table 6: Ablation study of the components in our framework on 20% labeled Synapse setting (IBSSL), MMWHS MR to CT setting (UDA) and 2% labeled 2% labeled M&Ms setting (SemiDG).

| Methods | A&D | w/o SVDA | w/o Diffusion | w/o DRS | w/o RS |
|---|---|---|---|---|---|
| IBSSL | 60.9 | 55.3 | 56.7 | 52.0 | 58.7 |
| UDA | 90.1 | 84.6 | 79.2 | 85.8 | 87.0 |
| SemiDG | 80.6 | 77.9 | 74.9 | 78.2 | 78.6 |

**Hyper-parameter Selection** We evaluate the performance of our method under different time step $t$ of the diffusion model and the number of sampled augmentations, as shown in Table 7.

Table 7: Ablation study of the noise step $t$ and the number of sampled augmentations on MMWHS CT to MR setting.

| Time step | 100 | 500 | 1000 | Aug # | 2 | 3 | 4 |
|---|---|---|---|---|---|---|---|
| Dice | 69.7 | 70.2 | **71.4** | Dice | 68.9 | **71.4** | 70.2 |

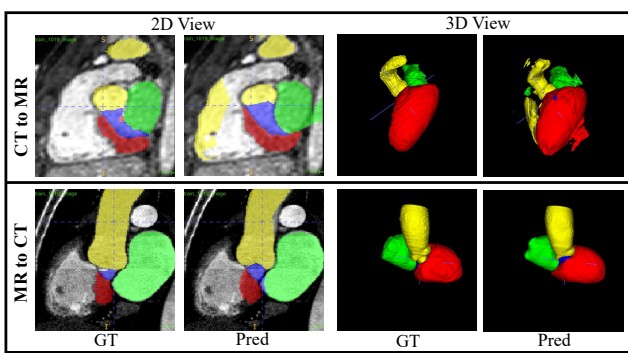

Figure 6: Visualization of the UDA task in 2D and 3D views.

**Visualization**   We also present the visualization results to further analyze our method. As shown in Figure 6, for the UDA task, our framework can generate smoother volumetric objects. Our framework also has detect minority classes, as can be seen in Appendix, which indicates the effectiveness of incorporating the class-imbalance awareness with the proposed difficulty-aware re-weighing strategy.

**Limitations**   The diffusion process introduces additional training costs; however, the inference efficiency remains unaffected as only the decoder trained using unlabeled data is utilized during inference. Moreover, the failure cases are mainly in the M&Ms dataset for SemiDG setting. Specifically, our method usually fails on the first and the last slices along the depth axis. Due to the restricted depth dimension (less than 10), the 2D slices and the corresponding masks vary significantly. In such a case, it is hard for our volumetric framework to capture depth-wise information for the first or last slice with only one neighboring slice as a reference, and thus leads to false positive results.

## 5   Conclusion

In this paper, we propose a generic framework for semi-supervised learning in volumetric medical image segmentation, called Aggregating & Decoupling. This framework addresses four related settings, namely SSL, class imbalanced SSL, UDA, and SemiDG. Specifically, the aggregating part of our framework utilizes a Diffusion encoder to construct a "common knowledge set" by extracting distribution-invariant features from aggregated information across multiple distributions/domains. On the other hand, the decoupling part involves three decoders that facilitate the training process by decoupling labeled and unlabeled data, thus mitigating overfitting to labeled data, specific domains, and classes. Experimental results validate the effectiveness of our proposed method under four settings.

The significance of this work lies in its ability to encourage semi-supervised medical image segmentation methods to address more complex real-world application scenarios, rather than just developing frameworks in ideal experimental environments. Furthermore, we have consolidated all four settings within a single codebase, enabling the execution of any task using a single bash file by merely adjusting the arguments. We believe that this consolidated codebase will be convenient for further research and beneficial for the community.

## Acknowledgement

We thank the anonymous NeurIPS reviewers for providing us with valuable feedback that greatly improved the quality of this paper! This work was supported in part by the Hong Kong Innovation and Technology Fund under Project ITS/030/21 and in part by Foshan HKUST Projects under Grants FSUST21-HKUST10E and FSUST21- HKUST11E and in part by the Project of Hetao Shenzhen-Hong Kong Science and Technology Innovation Cooperation Zone (HZQB-KCZYB-2020083).

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

# A  Appendix

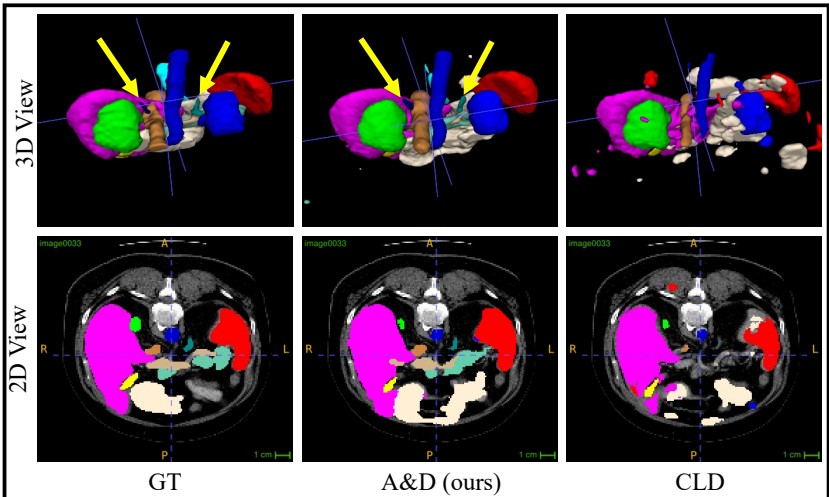

Figure 7: Visualization of the class imbalance SSL task in 2D and 3D views. The yellow arrows denote minority classes detected.

## A.1  More Details of Datasets and Implementation

The hyper-parameters for different datasets are shown in Table 8.

Table 8: Hyper-parameters for different datasets.

| Datasets | patch size | learning rate | batch size | feature size $F$ |
|---|---|---|---|---|
| LASeg | $112 \times 112 \times 80$ | 1e-2 | 4 | 32 |
| Synapse | $64 \times 128 \times 128$ | 3e-2 | 4 | 32 |
| MMWHS | $128 \times 128 \times 128$ | 5e-3 | 2 | 32 |
| M&Ms | $32 \times 128 \times 128$ | 1e-2 | 4 | 32 |

The details of the datasets and the pre-processing operations are as follows.

**LASeg Dataset for SSL**    The Atrial Segmentation Challenge (LASeg) dataset [72] provides 100 3D gadolinium-enhanced MR imaging scans (GE-MRIs) and LA segmentation masks for training and validation. Following previous work [8, 17], we split the 100 scans into 80 for training and 20 for evaluation. We use the processed datasets from [8] where all the scans were cropped centering at the heart region for better comparison of the segmentation performance of different methods and normalized as zero mean and unit variance. In the training stage, SS-Net [17] and BCP [76] use test set for validation to select the best model, which is unreasonable. We use labeled data instead and achieve better performances.

**Synapse Dataset for Class Imbalanced SSL**    The Synapse [73] dataset has 13 foreground classes, including spleen (Sp), right kidney (RK), left kidney (LK), gallbladder (Ga), esophagus (Es), liver(Li), stomach(St), aorta (Ao), inferior vena cava (IVC), portal & splenic veins (PSV), pancreas (Pa), right adrenal gland (RAG), left adrenal gland (LAG) with one background and 30 axial contrast-enhanced abdominal CT scans. We randomly split them as 20,4 and 6 scans for training, validation, and testing, respectively. Following DHC [25], we run the experiments 3 times with different random seeds.

**MMWHS Dataset for UDA**    Multi-Modality Whole Heart Segmentation Challenge 2017 dataset (MMWHS) [74] is a cardiac segmentation dataset including two modality images (MR and CT). Each modality contains 20 volumes collected from different sites, and no pair relationship exists between modalities. Following the previous work [41], we choose four classes of cardiac structures. They are the ascending aorta (AA), the left atrium blood cavity (LAC), the left ventricle blood cavity (LVC), and the myocardium of the left ventricle (MYO). For the pre-processing, follow [41], (1) all the scans

were cropped centering at the heart region, with four cardiac substructures selected for segmentation; (2) for each 3D cropped image top 2% of its intensity histogram was cut off for alleviating artifacts; (3) each 3D image was then normalized to [0, 1]. To make a fair comparison, we keep the test set the same with prior arts [38, 41, 44].

**M&Ms Dataset for SemiDG**  The multi-center, multi-vendor & multi-disease cardiac image segmentation (M&Ms) dataset [75] contains 320 subjects, which are scanned at six clinical centers in three different countries by using four different magnetic resonance scanner vendors, i.e., Siemens, Philips, GE, and Canon. We consider the subjects scanned from different vendors are from different domains (95 in domain A, 125 in domain B, 50 in domain C, and another 50 in domain D). We use each three of them as the source domain for training and the rest as the unseen domain for testing. For the pre-processing, (1) all the scans were cropped with four cardiac substructures selected for segmentation; (2) for each 3D cropped image top and bottom 0.5% of its intensity histogram was cut off for alleviating artifacts; (3) each 3D image was then normalized to [0, 1]. Since the data has very few slices on the z-axis (less than 16), the previous work used 2D-based solutions. In this work, since we aim to design a generic framework for volumetric medical image segmentation, we stacked the z-axis to 32 to meet the minor requirement for our encoder with four down-sampling layers. This dataset can also be considered as the extreme case of 3D segmentation tasks.

Table 9: Comparison of computational costs of state-of-the-art methods on different settings with the performances in terms of Dice score.

| Methods | Param.(M) | Inference time (s/iter) | SSL | UDA |
|---|---|---|---|---|
| SS-Net [17] | 75.672 | 21.20 | 88.55 | 78.2 |
| CLD [15] | 75.554 | 39.58 | 85.37 | 75.4 |
| vMFNet [55] | 20.423 | 4.89 | 73.88 | 72.3 |
| EPL [54] | 80.939 | 5.52 | 76.49 | 71.9 |
| A&D (ours) | 57.894 | 3.69 | 90.31 | 90.1 |

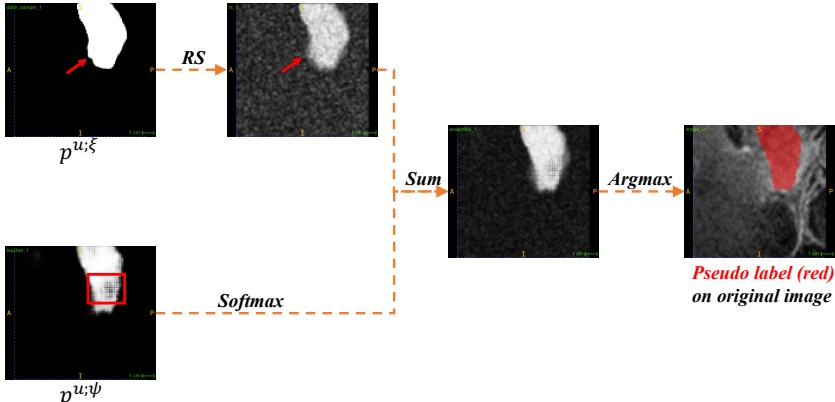

Figure 8: Visualization of the RS process of the foreground class on LASeg dataset. The probability map $p^{u;\psi}$ of the difficulty-aware decoder may have low confidence in the inner region (red box), whereas the probability map $p^{u;\xi}$ of the diffusion decoder may have inaccurate boundaries but with very high confidence (red arrows).

## A.2 More Analyses

**Comparison of Computational Costs**  We compared the parameters and the inference time of our method with the most SOTA methods in different settings: SS-Net [17] on LASeg dataset for SSL, CLD [15] on Synapse dataset for IBSSL, and vMFNet [55] as well as EPL [54] on M&Ms dataset for SemiDG, as shown in Table 9. From the table we can observe that our method has the fewest parameters except for vMFNet. Although vMFNet has only 20 M parameters, their performances (73.88% and 72.3%) on SSL and UDA tasks are significantly lower than ours (89.40% and 90.0%).

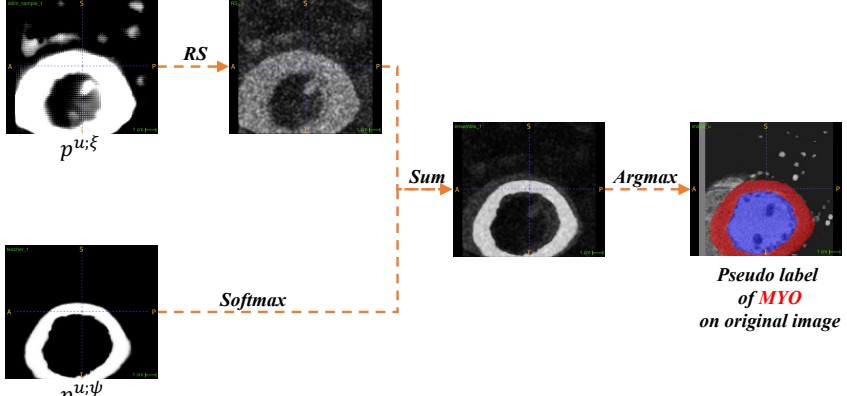

Figure 9: Visualization of the RS process of the myocardium of the left ventricle (MYO) class which is the class with worst performance on MR to CT setting of MMWHS dataset. In this case, the probability map $p^{u;\xi}$ of the diffusion decoder contains more error regions due to the ambiguous boundaries and noise but also with very high confidence.

As for the inference time, we can observe that our A&D is the fastest. This can be attributed to our effective aggregating and decoupling strategies, enabling our method to exclusively utilize the unlabeled branch for inference.

**Visualization of the Re-parameterize & Smooth (RS)**   As shown in Figure 8, the output probability map $p^{u;\xi}$ of diffusion with DDIM $D(\xi)$ is with very high confidence with its prediction, however, the results are not stable since the unlabeled data is *unseen* during the training process of the diffusion decoder, especially for some problematic classes (MYO of MMWHS, Figure 9) with ambiguous boundaries and noise. Thus, if we sum it with the map $p^{u;\psi}$ generated by the V-Net decoder $D(\psi)$, the error regions (e.g., upper right corner) with high confidence will surpass some correct regions of $p^{u;\psi}$ with lower confidence and further harm the quality of the final pseudo label. Moreover, in some cases, the two output probability maps have complementary properties (Figure 8), indicating the effectiveness of ensembling them for the high-quality pseudo labels.

**Ablation on the Effectiveness of Decoupling the Labeled and Unlabeled Data Training Flows**
Based on our final framework, we add an additional training process with labeled data on the decoder $D(x^u; \theta)$ trained with unlabeled data to verify the effectiveness of the decoupling idea. Compared with the final A&D framework, when adding an additional labeled data training branch, the performance in terms of Dice drops from 90.03% to 86.94% on the MR to CT setting of the MMWHS dataset. The result indicates that when the predictor is trained with labeled and unlabeled data, it may get over-fitted to the easier labeled data flow, which verifies the effectiveness of the key idea of our decoupling stage.

