# OpenReview forum: "Towards Generic Semi-Supervised Framework for Volumetric Medical Image Segmentation"
_NeurIPS.cc/2023/Conference — NeurIPS 2023 poster_

### Official Review · Reviewer_GWMf · 2023-06-27

**Soundness:** 3 good
**Presentation:** 3 good
**Contribution:** 3 good
**Rating:** 7
**Confidence:** 4

**Summary:**

This paper tackles semi-supervised medical image segmentation over multiple domains, generalizing unsupervised domain adaptation and semi-supervised learning. A multiple branch encoder-decoder architecture with a shared encoder is proposed. The first branch is a diffusion model with V-Net architecture trained over labeled images and their noisy labels, where the encoder aims to extract domain-invariant features. The second decoder is trained over labeled images to combat class-imbalance by reweighting the loss with respect to Population Stability Index. The third branch is trained over unlabeled images with pseudolabels, where pseudolabels are ensembles of probability maps predicted by the other two decoders. To account for the sparse predictions of diffusion model decoder, its probability predictions are reparamatrized via Gumbel-Softmax prior to ensembling into pseudolabels. Each branch is trained over a combination of cross-entropy and dice loss, with input images and labels varying with respect to the branch.

**Strengths:**

Adopting diffusion models for generalized semi-supervised segmentation and domain adaptation is novel. Abundant experiments over multiple tasks and datasets demonstrate the benefit of the proposed method, particularly for unsupervised domain adaptation.


**Weaknesses:**

Unsupervised/semi-supervised segmentation and domain adaptation in medical imaging are well-explored domains with several missing related works:

Gu et al. “ConFUDA: Contrastive Fewshot Unsupervised Domain Adaptation for Medical Image Segmentation”, 2022

Xia et al. “Uncertainty-aware multi-view co-training for semi-supervised medical image segmentation and domain adaptation”, 2020

Bian et al. “DDA-Net: Unsupervised cross-modality medical image segmentation via dual domain adaptation”, 2021

Perone et al. “Unsupervised domain adaptation for medical imaging segmentation with self-ensembling”, 2019

Shin et al. “COSMOS: Cross-Modality Unsupervised Domain Adaptation for 3D Medical Image Segmentation based on Target-aware Domain Translation and Iterative Self-Training”, 2022

Bateson et al. “Source-Relaxed Domain Adaptation for Image Segmentation”, 2021

Liu et al. “S-CUDA: Self-cleansing unsupervised domain adaptation for medical image segmentation”, 2021

Liu et al. “ACT: Semi-supervised Domain-adaptive Medical Image Segmentation with Asymmetric Co-Training”, 2022

Particularly, Liu et al. 2022 propose decoupled semi-supervised training and domain adaptation branches, a similar motivation to the proposed method. The authors should highlight their novelty against these methods that are not discussed.


**Questions:**

None

**Limitations:**

Limitations are discussed

---

> ### Author Rebuttal · Authors · 2023-08-10
>
> Sincere gratitude to reviewer GWMf for the recognition of our idea and suggestion for supplementing the related work. We hope our answers can address all your concerns. We are looking forward to having further discussions with you.
>
> Regarding the differences with ACT:
>
> (1) **The settings are different.** ACT is tailored specifically for Semi-supervised Domain Adaptation, which is a simpler setting than Unsupervised Domain Adaptation since some labeled data from the target domain is available for training. Our goal is to build a generic framework that unifies four settings, including UDA. Furthermore, our A&D can also solve the SemiDA setting.
>
> (2) **The motivation for decoupling and the data we want to decouple are different.** ACT decouples the semi-supervised training and domain adaptation branches based on the observation that **abundant labeled source data** can easily dominate training when directly combined with **limited labeled target data**. However, we found that the **limited labeled data** can dominate training when combined with **abundant unlabeled data**, hence we decouple them to avoid the unsupervised branch overfit with the labeled data. Experiments in Figure 3 and Appendix A.2 support and verify of our motivation.
>
> In terms of the **novelty** compared to previous works, our work is the first to unify SSL, Imbalanced SSL, UDA, and SemiDG. This extension not only amplifies the scope and versatility of SSL-based frameworks in medical image segmentation but also stands in stark contrast to earlier approaches that remained confined to singular domains such as SSL or UDA.
>
> The significance of this work lies in its ability to encourage semi-supervised medical image segmentation methods to address more complex real-world application scenarios, rather than just developing frameworks in ideal experimental environments. Furthermore, we have consolidated all four settings within a single codebase, enabling the execution of any task using a single bash file by merely adjusting the arguments. We believe that this consolidated codebase will be convenient for further research and beneficial for the community.

---

> > ### Comment · Reviewer_GWMf · 2023-08-14
> >
> > I have read author rebuttals and discussions with all reviewers. While it seems that the original scores are generally borderline, I believe that the authors addressed most concerns regarding novelty and motivation, particularly specific to medical imaging. Thus, I keep my original score.
> >
> > Thank you,

---

### Official Review · Reviewer_HPgc · 2023-07-03

**Soundness:** 3 good
**Presentation:** 2 fair
**Contribution:** 3 good
**Rating:** 4
**Confidence:** 5

**Summary:**

This paper addressed both the semi-supervised issue and the domain adaptation issue for medical image segmentation. The proposed method incorporated the diffusion model and used a consistency-based regularization for training. Performance on several public datasets is good for the segmentation tasks.

**Strengths:**

1. Applying a diffusion model to reduce the domain gaps is reasonable and achieves performance gains according to the results.
2. Extensive experiments and sufficient comparisons.
3. The motivation is sound. Considering both tasks together is now the research attention.

**Weaknesses:**

1. There are some recent papers considering the two tasks as well. See:
[1] Bai Y, Chen D, Li Q, et al. Bidirectional Copy-Paste for Semi-Supervised Medical Image Segmentation[C]//Proceedings of the IEEE/CVF Conference on Computer Vision and Pattern Recognition. 2023: 11514-11524.
[2] Wu H, Li X, Lin Y, et al. Compete to Win: Enhancing Pseudo Labels for Barely-supervised Medical Image Segmentation[J]. IEEE Transactions on Medical Imaging, 2023.
Therefore, the problem definition is new.

2. It's a bit over-claimed to unify four tasks together. For example, this paper does not provide unique designs for imbalanced learning. Do not make the claims too broad.
3. The performance is not SOTA, especially on the popular LA dataset, see [1]. Moreover, about the comparison, it is weird that there are different compared methods for different tasks.
4. Regarding the main task (semi-supervised learning), the technical insight is not clear. The domain gap should be a critical issue here. Please elaborate more on the take-home insights rather than tricks.

**Questions:**

See the above weaknesses

**Limitations:**

See the above weaknesses

---

> ### Author Rebuttal · Authors · 2023-08-10
>
> Sincere gratitude to reviewer HPgc. We hope our answers can address all your concerns. We are looking forward to having further discussions with you.
>
> **Q1: There are some recent papers considering the two tasks as well ...**
>
> We believe there might be some misunderstandings regarding the differences between the mentioned papers and our own work. The papers the reviewer mentioned utilized datasets like the ACDC dataset, Pancreas-NIH dataset, Colon cancer segmentation dataset, and LA dataset—all originating from a consistent source domain. As a result, their focus was confined to situations without any domain shift, i.e., "semi-supervised setting." However, our work is distinctive in that it can achieve the best performance on four distinct settings simultaneously: SSL, Imbalanced SSL, UDA, and SemiDG. This expansive coverage underscores our ability to address a broader range of challenges. As such, our problem formulation and methodology stand as a novel contribution.
>
> **Q2: It's a bit over-claimed to unify four tasks together. For example, this paper does not provide unique designs for imbalanced learning. Do not make the claims too broad.**
>
> Indeed, we introduce a novel approach to address imbalanced learning - our Difficulty-aware Re-weighting Strategy (DRS). Moreover, our proposed method demonstrates a notable superiority over prior state-of-the-art solutions within the class-imbalanced SSL settings, as clearly illustrated in Table 1. Based on these achievements, we are confident that our approach effectively unifies the Imbalanced SSL setting.
>
> **Q3: The performance is not SOTA, especially on the popular LA dataset, see [1]. Moreover, about the comparison, it is weird that there are different compared methods for different tasks.**
>
> (1) The results in [1] reveal Dice values of 88.02 and 89.62 for LA datasets with 5% and 10% labeled data, respectively. While these values do show a slight improvement over our method's performance, which records scores of 86.95 and 89.40 in the same scenarios, it's important to note that our approach is designed to cater to broader scenarios. Specifically, our method addresses not only the SSL setting, as [1] does, but also encompasses imbalanced SSL, UDA, and SemiDG settings, making it more versatile and applicable across various challenges.
>
> (2) Different compared methods for different tasks is due to that we conducted fair comparisons with existing state-of-the-art methods in the SSL, UDA, and SemiDG benchmarks. Currently, these three settings and their corresponding methods were individually formulated and designed, leading to the diversity in the compared approaches. However, it's worth noting that our method consistently outperforms the best-performing methods tailored to these three settings. Our approach consistently achieves superior performance across all three settings, demonstrating its efficacy and generalization than other existing STOA methods.
>
> **Q4: Regarding the main task (semi-supervised learning), the technical insight is not clear. The domain gap should be a critical issue here. Please elaborate more on the take-home insights rather than tricks.**
>
> Regarding the semi-supervised learning task, our key technical contribution pertains to the identification of the limited impact of unlabeled data in the training process of the existing SSL frameworks, which is due to the unlabeled data easily getting overwhelmed by the labeled data. Aiming to solve this issue, we design the Aggregating & Decoupling framework to decouple the labeled data and the unlabeled data training branches. The motivation is mainly illustrated in Figure 3, and the validation of the decoupling process is demonstrated in the Appendix.

---

> > ### Comment · Reviewer_HPgc · 2023-08-14
> >
> > Thanks for the author's response. It has addressed most of my concerns. However, a few remaining issues persist:
> >
> > 1. Regarding the performance of individual tasks e.g., [1] in SSL.
> > While I understand that the author's response considers the simultaneous impact of domain gap and semi-supervised segmentation, I would like to highlight that [1] also takes into account these factors. We can see that the domain gap is more severe with scarce labeled data. Combining the two tasks together is not new in related research. Furthermore, the performance of [1] is better.
> >
> > 2. Unifying four tasks for broader scenarios.
> > I do appreciate the initiative to unify four tasks since semi-supervised segmentation for real applications would be limited by various challenges, imbalance issues, domain gaps, SSL, etc. However, I would like to discuss the motivation again.
> > (1) the large foundation model would be a more suitable choice now, compared to the proposed unified model.
> > (2) when we apply the model to real scenarios, do the various challenges exist together? Meanwhile, we can also leverage some engineering tricks to relieve the issues here. For example, producing a more balanced dataset, focusing on one task first...
> >
> > Overall, I like the responses but there are still some critical concerns for me.

---

> > > ### Author Response · Authors · 2023-08-14
> > >
> > > We thank reviewer HPgc for the reply and the suggestions.
> > >
> > > **Issue 1.**
> > > First, we need to clarify that the **domain gaps** in medical images lie in different machines, hospitals, and modalities. Specifically, the subjects in M&Ms dataset are scanned at **six clinical centers** in **three different countries** by using **four different magnetic resonance scanner vendors**, i.e., Siemens, Philips, GE, and Canon. Subjects from different domains show significant distinctions in shape, texture, density and etc., as shown in Figure 1 in [2]. However, in LASeg dataset used in [1], clinical images were acquired solely with the Siemens scanner from the same hospital.
> > >
> > > Moreover, the average similarity score among all pairs of samples in LASeg is **0.73171**, while the score between different modalities in MMWHS dataset for UDA is **0.20397**, and between different domains in M&Ms dataset for SemiDG is **0.32752**.
> > > Thus, the problem definition in [1] is **not the domain gap issue**; they also use **distribution gap** instead to distinguish.
> > >
> > > Furthermore, we also tested the performance of BCP [1] in the MR to CT setting of UDA, which is with the largest domain gap since the images in source and target domains are with different modalities. BCP only obtained a 62.1% Dice score, while ours is 90.0%, which also verifies the distinctions of our targeted issue and that in [1]. **Hence, our setting and problem definition are new.**
> > >
> > > **Regarding the performance**, BCP [1] and SS-Net **used the test set for validating in their training stages to select the best models** (as can be found in the GitHub repo of BCP: Line 175 in code/LA_BCP_train.py and Line 22 in code/utils/test_3d_patch.py), hence we believe there exists **data-leakage problem** and the results are not correct. In contrast, the results in our paper are obtained strictly following UA-MT [3] where the test set is unseen in the training stage. Furthermore, we show below the performance of our A&D under different validation settings to make a fair comparison with BCP and SS-Net:
> > >
> > > || 5% labeled LA| 10% labeled LA|
> > > |--|--|--|
> > > |SS-Net (test data for val)|86.33|88.55|
> > > |BCP (test data for val)|88.02|89.62|
> > > |Ours (no validation, reported in paper)|86.95|89.40|
> > > |Ours (labeled data for val)|**89.27**|**90.50**|
> > > |Ours (test data for val)|89.74|90.69|
> > >
> > > As can be seen, when using the test set for selecting the model the same as BCP [1] and SS-Net, our method outperforms BCP. Additionally, when using **labeled data for validation**, our method also outperforms BCP and SS-Net. We will regard these results as our final results since they are correct with no data-leakage problem while achieving better performances.
> > >
> > >
> > >
> > > **Issue 2 (1)** We agree with the reviewer that a large foundation model would be more suitable. However, the current results using foundation models, particularly SAM (segment anything model), in a zero-shot manner are far below state-of-the-art methods [4].
> > > Some approaches [5] that achieve results on par with SOTA methods do so by fine-tuning the foundation model, but this results in a long running time. In clinical applications, accuracy on a specific segmentation task, such as abdominal multi-organ segmentation, is more important than the versatility of segmenting different targets.
> > >
> > > **(2)** Domain gap issues are common along with SSL issues when applying SSMIS methods to real-world application scenarios. One common way to apply such a method is to integrate it into medical image analysis devices that are sold to different hospitals. Each hospital usually has limited labeled data and abundant unlabeled data scanned with different scanners, resulting in significant variation between data samples from different hospitals, machines, and even modalities.
> > > The distribution gap issue in the SSL setting and the domain gap issue in UDA and SemiDG settings share similarities but are not jointly considered by the prior works.
> > >
> > > Furthermore, it is not practical to obtain a **balanced dataset**, due to the anatomical structures of medical segmentation targets, which are naturally imbalanced within **each sample**. Unlike in the natural image domain, we cannot simply increase the number of samples for minority classes to obtain a balanced dataset.
> > >
> > >
> > > [2] Yao et al. "Enhancing pseudo label quality for semi-supervised domain-generalized medical image segmentation." _AAAI_, 2022.
> > >
> > > [3] Yu et al. "Uncertainty-aware self-ensembling model for semi-supervised 3D left atrium segmentation." _MICCAI_, 2019.
> > >
> > > [4] Roy et al. "Sam. md: Zero-shot medical image segmentation capabilities of the segment anything model." _arXiv_ (2023).
> > >
> > > [5] Gong et al. "3DSAM-adapter: Holistic Adaptation of SAM from 2D to 3D for Promptable Medical Image Segmentation." _arXiv_ (2023).

---

> > > > ### Comment · Reviewer_HPgc · 2023-08-15
> > > >
> > > > Thanks for the thoughtful responses. One more question here (If time allows):
> > > > As far as I know, both BCP and SS-Net focused on semi-supervised segmentation when trained with extremely limited labels. Thus, the differences between the limited labeled data and the rest unlabeled samples are significant. This issue might be not severe on the LASeg dataset. But on another experimental dataset (ACDC), the experiments of BCP demonstrated that the issue significantly limited the performance so they would like to align different domains (or distributions, all good). If there are no unique designs of the proposed model for 3D segmentation, an additional experiment is needed to show the performance on ACDC. Furthermore, the ACDC dataset is split into fixed training, validation, and testing sets.
> > > >
> > > > Overall, the responses are sufficient and the authors have done great efforts for improving this manuscript. These discussions should be added in the final version.

---

> > > > > ### Comment · Reviewer_HPgc · 2023-08-19
> > > > >
> > > > > Given the technical novelty and plausible contributions, I prefer to maintain my original score.
> > > > >
> > > > > To further improve this paper, the authors are suggested to highlight one or two major technical contributions, refraining from making overly big claims.  The current framework combines many modules and looks a bit incremental. Meanwhile, the results on the ACDC dataset should be provided since it involves tackling two studied problems concurrently.
> > > > >
> > > > > Furthermore, NeurIPS might not be the best venue for this work and other medical-related Conferences or Journals should be more suitable.

---

> > > > > > ### Author Response · Authors · 2023-08-19
> > > > > >
> > > > > > Thanks for the valuable suggestions of reviewer HPgc on improving our work.
> > > > > >
> > > > > > Regarding the ACDC dataset. **Again, the main issue of the ACDC dataset is different from our UDA and SemiDG settings.** The samples of ACDC were obtained using MRI scanners of the same magnetic resonance scanner vendors (Siemens) from the same hospital (University Hospital of Dijon). **Thus, there is no domain gap within the ACDC dataset, and even the distribution gap, i.e., the differences between the limited labeled data and the rest unlabeled samples, stated in [1] is not clear (with an average cosine similarity of 0.80134 between every pair of data).**
> > > > > >
> > > > > > Moreover, we also tested our method on the 5% labeled ACDC dataset, the results are shown as follows:
> > > > > >
> > > > > > | | Exp 1| Exp2| Exp 3| Mean|
> > > > > > |--|--|--|--|--|
> > > > > > |BCP | 88.10 | 77.55| 79.48| **81.71**|
> > > > > > |BCP w/o pre-training |82.35|71.03| 74.31 |**75.90**|
> > > > > > |A&D (ours)|77.29 | 78.65 | 75.20 | **77.04**|
> > > > > >
> > > > > > The results reported in [1] are almost too good which may be due to over-fitted to the selected labeled data, so we reproduced the experiment (Exp 1 is the same setting as [1], our reproduced results are even higher than the reported) with different labeled data, **the performances dropped and varied significantly**.
> > > > > > In our device, the BCP involves 1.5h pre-training and 3.5h self-training, while ours using only 1h can obtain results better than "BCP w/o pre-training".
> > > > > >
> > > > > > Notably, the ACDC dataset is very similar to **one domain** of the M&Ms dataset for SemiDG in our paper, while the M&Ms dataset is closer to the clinical scenarios.
> > > > > >
> > > > > > | | results of domain A of M&Ms| results of ACDC test set|
> > > > > > |--|--|--|
> > > > > > |BCP (trained on 5% labeled ACDC) | **37.27** | 87.71|
> > > > > > |BCP (trained on 2% labeled M&Ms) |45.68|**42.83**|
> > > > > > |Ours (trained on 5% labeled ACDC) | 67.35 | 77.29|
> > > > > > |Ours (trained on 2% labeled M&Ms) |75.52|73.10|
> > > > > >
> > > > > > BCP has significant performance drops when **tested on unseen domains** which is also the **actual domain gap problem** (boldfaced in the table), while our A&D shows great generalizability.
> > > > > >
> > > > > >
> > > > > >
> > > > > > **Regarding the main contributions**, our work is the first to unify SSL, Imbalanced SSL, UDA, and SemiDG settings by aggregating and decoupling the diffusion model for solving main obstacles to unification, along with several strategies for further improving the generalizability. This extension not only amplifies the scope and versatility of SSL-based frameworks in medical image segmentation but also stands in stark contrast to earlier approaches that remained confined to singular domains such as SSL or UDA. There is no previous work noticing the similarity of these scenarios, hence **our contribution is not incremental**. **We believe our work can encourage semi-supervised medical image segmentation methods to address more complex real-world application scenarios, rather than just developing frameworks in ideal experimental environments.**

---

### Official Review · Reviewer_GWTw · 2023-07-06

**Soundness:** 3 good
**Presentation:** 3 good
**Contribution:** 3 good
**Rating:** 4
**Confidence:** 5

**Summary:**

This work proposes a generic framework for semi-supervised learning (SSL) in medical image segmentation. They identify two obstacles that prevent the applicability of SSL in real-world scenarios. Then they propose Aggregating & Decoupling to alleviate the issues under several settings, including class imbalanced SSL, UDA, and SemiDG. They show the proposed method can mitigate overfitting to labeled data, specific domains, and classes across four settings and many datasets.

**Strengths:**

The paper is well-structured and easy to follow. The proposed method is generalizable to four different settings, enhancing the applicability of SSL in real-world scenarios. The presentation is very clear, especially the introduction, where the obstacles, and motivations are clearly stated.

**Weaknesses:**

My biggest concern about this work is the effectiveness in more generic applications, e.g., standard computer vision benchmarks. I appreciate the authors identifying obstacles that hurt the performance of SSL in real-world applications. However, what is the relationship between the proposed approach and medical image segmentation? Is any proposed component tailored to deal with the issues with medical images? In my eye, the proposed framework is generic and should be working widely well in many applications instead of only focusing on medical images. It would be great if the authors could verify the effectiveness of the proposed methods in CV benchmarks instead of only medical imaging applications if the proposed method is not designed or tailored to medical settings. I would consider re-rating if the authors response to my concern.

**Questions:**

See weakness.

**Limitations:**

Yes.

---

> ### Author Rebuttal · Authors · 2023-08-10
>
> Sincere gratitude to reviewer GWTw for the recognition of our idea and suggestion for extending the application scenarios to natural images. We hope our answers can address all your concerns. We are looking forward to having further discussions with you.
>
> **Weaknesses: My biggest concern about this work is the effectiveness in more generic applications ...**
>
> We thank the reviewer for this insightful comment.
>
> **Regarding the relationship of our method with medical image segmentation:**
>
> (1) Our DRS is tailored for the imbalance issue in medical images. In volumetric medical image segmentation, the MRI/CT scans are obtained by scanning the same region of the patients, hence the anatomical structure and segmentation targets within each dataset the similar character but vary in size and texture. **The imbalance issue lies in organs with tiny sizes, which can be regarded as local imbalance.** In contrast, natural images are diverse and from complex scenarios, the objects with each image, **the imbalance issue lies in the objects with scarce samples, which can be regarded as global imbalance**. DRS computes the imbalance ratio within **each training sample**, thus, is tailored for the local imbalance issue in medical images.
>
> (2) The evaluation of our method is in a patient-by-patient manner, the failure to segment a target has larger impacts on the overall results, unlike the class-by-class approach commonly applied to natural images. Therefore, our method places a stronger emphasis on accurately handling small objects. This design is tailored for the clinical settings, where even minor mis-segmentation of organs and objects can exert a significant impact on the overall results.
>
> (3) Our framework, particularly the aggregating stage of the diffusion model, serves as a robust backbone for the volumetric medical image segmentation. Powerful pre-trained feature extractors is unavailable in SSVMIS domain due to the limited data. Observed the failure of existing 3D encoders for extracting domain-invariant features, we propose to aggregate the multi-domain information within both scarce labeled and unlabeled data into one Diffusion-VNet encoder to solve this issue.
>
> (4) Our SVDA comprises a combination of 3D spatial- and voxel-wise transformations which are tailored for the gray-scale MRI/CT volumetric data, in contrast with the 2D augmentations for RGB natural images.
>
> (5) Our aggregating and decoupling strategies for diffusion are tailored for the limited medial data compared with large amounts of natural images. In natural image datasets, both the labeled and unlabeled data are abundant, hence the observation in Figure 3 is uncommon. However, in medical image datasets, even the unlabeled data is scarce, leading to over-fitting problem of the labeled data. We decouple the training of labeled and unlabeled data to solve this issue.
>
> **Regarding the reason why our method can’t be applied to CV:**
>
> (1) Our DRS computes the imbalance ratio in a **sample-by-sample manner** to solve the aforementioned local imbalance issue in medical image. In the natural image domain, the classes may be quite balanced in one sample but seriously imbalanced across the whole training set. Thus, applying our DRS to natural images will lead to incorrect representations of the class-wise difficulty.
>
> (2) When applied to CV, our perception-based diffusion model is hard to pre-train since it also takes the masks as inputs. Thus, it is hard to align with the SOTA approaches equipped with powerful pre-trained backbones in the natural image domain.
>
> (3) Our SVDA is tailored for the gray-scale volumetric data, which is unsuitable for the RGB 2D data in natural images.
>
> (4) Notably, we have evaluated several SOTA CV semi-supervised semantic segmentation methods (CPS and GTA [3]) on our tasks but obtained inferior results, as shown in the table below.
>
>
> | Method | SSL | IBSSL |
> | --| -- | --|
> | CPS (CVPR'21) | 87.91 | 37.4 |
> | GTA (NeurIPS'22) | 88.25 | 39.3|
> | A\&D (ours) | 89.40| 60.2 |
>
> Moreover, directly extending existing CV semi-supervised classification methods to segmentation with CPS as the baseline is also not effective, as shown in Table 1 of the paper.
>
> [1] Fan, Jiashuo, et al. "Ucc: Uncertainty guided cross-head co-training for semi-supervised semantic segmentation." CVPR, 2022.
> [2] Jin, Ying, Jiaqi Wang, and Dahua Lin. "Semi-supervised semantic segmentation via gentle teaching assistant." Advances in Neural Information Processing Systems 35 (2022): 2803-2816.

---

> > ### Comment · Area_Chair_9rnJ · 2023-08-18
> > **Rebuttal to GWTw**
> >
> > Dear GWTw
> >
> > Could you have a look at the rebuttal to see if your questions have been clarified?
> >
> > Thanks,
> > Your AC

---

> > ### Comment · Reviewer_GWTw · 2023-08-18
> > **Response to Rebuttal**
> >
> > I have carefully read the response. CV tasks can also be "sample-by-sample" and have imbalanced issues. I still feel there is no strong relationship between the proposed methods and the medical domain when I reread the full paper. Therefore, I keep my original score.

---

> > > ### Author Response · Authors · 2023-08-19
> > >
> > > We respectfully disagree with the notion that the "sample-by-sample" and imbalanced issues are the same in CV and medical images. To illustrate this point, let's consider a semi-supervised setting in CV that features a small-sized cat and a larger-sized person [1]. In this case, the cat represents the minority class and should be assigned higher weights, as demonstrated in GTA-Seg [1], which employs a pixel-wise re-weighting strategy. However, if we were to apply our method directly in this scenario, which relies on accumulating samples with PSI (Population Stability Index), it could lead to an inaccurate estimation of the "cat" class. The reason for this is that our method would consider the cat as the majority class across the entire dataset, rather than recognizing it as the minority class within that specific image. As a result, the weight assigned to the cat class would decrease, which is not desirable. In contrast, when applying our method to medical image scenarios characterized by severe imbalance issues within each individual sample, the cumulative approach proves highly advantageous. By aggregating information from multiple samples, our method significantly enhances the robustness and precision of the imbalance ratio estimation. The cumulative nature of our method effectively tackles the imbalanced nature of medical images, leading to outcomes that are more accurate and reliable.
> > >
> > > Furthermore, the pixel-wise re-weighting strategy is also not proper for the medical domain since the class distribution in one image is already serve imbalanced (> 500:1), see the table in our rebuttal. Only masking out more than 90% of pixels of the majority class can obtain a balanced distribution but will result in training difficulties of the majority classes.
> > >
> > > It will be our future work to design a generic framework for both natural and medical image domains. In this work, our primary focus lies in enhancing semi-supervised medical image segmentation methods to effectively address complex real-world application scenarios, rather than solely dedicating our efforts to developing frameworks for semi-supervised learning, which carries strong assumptions.
> > >
> > > [1] Jin, Ying, Jiaqi Wang, and Dahua Lin. "Semi-supervised semantic segmentation via gentle teaching assistant." Advances in Neural Information Processing Systems 35 (2022): 2803-2816.

---

### Official Review · Reviewer_yd6u · 2023-07-06

**Soundness:** 3 good
**Presentation:** 3 good
**Contribution:** 3 good
**Rating:** 5
**Confidence:** 3

**Summary:**

This paper proposes an Aggregating & Decoupling framework, which unifies the SSL, Class Imbalanced SSL, UDA, and SemiDG for volumetric medical image segmentation with one generic framework.

**Strengths:**

This paper explores a generic framework for the SSL, Class Imbalanced SSL, UDA, and SemiDG of volumetric medical image segmentation.

**Weaknesses:**

The description of the method is not clear.

**Questions:**

1. How are the running time and parameters of the proposed method compared to other methods?
2. The proposed approach appears to require separate training for different tasks, and whether it is possible to train only once for the SSL, Class Imbalanced SSL, UDA, and SemiDG, and truly achieve a generic framework.
3. Do ablation experiments on other datasets also demonstrate the performance of the proposed architecture and components?

**Limitations:**

The authors have analyzed the limitations of the proposed method.

---

> ### Author Rebuttal · Authors · 2023-08-10
>
> Sincere gratitude to reviewer yd6u for the recognition of our idea. We hope our answers can address all your concerns. We are looking forward to having further discussions with you.
>
> **Q1: How are the running time** **and parameters of the proposed method compared to other methods?**
>
> We compared the parameters and the inference time of our method with the most SOTA methods in different settings: SS-Net on LASeg dataset for SSL, CLD on Synapse dataset for IBSSL, and vMFNet as well as EPL_SemiDG on M&Ms dataset for SemiDG:
>
> | Methods | Param.(M) | Inference time (s/iter) | SSL| UDA|
> | --| -- | --| --| -- |
> | SS-Net | 75.672 |21.20| 88.55 | 78.2 |
> | CLD | 75.554 | 39.58 | 85.37| 75.4|
> | vMFNet | 20.423 | 4.89 | 73.88 | 72.3 |
> | EPL_SemiDG | 80.939 | 5.52 | 76.49 | 71.9|
> | A\&D (ours) | 57.894 | 3.69 | 89.40 | 90.0 |
>
> From the table we can observe that our method has the fewest parameters except for vMFNet. Although vMFNet has only 20 M parameters, their performances (73.88% and 72.3%) on SSL and UDA tasks are significantly lower than ours (89.40% and 90.0%).
>
> As for the inference time, we can observe that our A&D is the fastest. This can be attributed to our effective aggregating and decoupling strategies, enabling our method to exclusively utilize the unlabeled branch for inference.
>
>
> **Q2: The proposed approach appears to require separate training for different tasks, and whether it is possible to train only once for the SSL, Class Imbalanced SSL, UDA, and SemiDG, and truly achieve a generic framework.**
>
> We appreciate the reviewer's suggestion regarding training the model once and testing it across SSL, UDA, and SemiDG settings. Undoubtedly, this concept holds great value. However, its viability hinges on the presence of a uniform segmentation task across these settings, implying that the same target must be segmented in SSL, UDA, and SemiDG scenarios. However, our present paper employs widely-used benchmark datasets, which were chosen to ensure a fair comparison with existing state-of-the-art SSL, class-imbalanced SSL, UDA, and SemiDG methods, respectively. It's crucial to know that these benchmark datasets do not share a consistent segmentation objective. Specifically, prior best-performing SSL methods employs the LASeg dataset for benchmarking left atrium segmentation, the Class Imbalanced SSL methods employs the Synapse dataset to segment 13 abdominal organs, and prior best-performing UDA methods employ the MMWHS dataset to segment 3 cardiac structures. Consequently, the inconsistent segmentation objectives within these datasets render the implementation of the aforementioned idea unfeasible.
>
> To achieve the aforementioned idea, we will endeavor to collect an extensive dataset from various hospitals and conduct a comprehensive and equitable assessment of the top-performing SSL, Imbalanced SSL, UDA, and SemiDG methods in the new benchmark.
>
> **Q3: Do ablation experiments on other datasets also demonstrate the performance of the proposed architecture and components?**
>
> We add additional ablation experiments on 20% labeled Synapse and 5% labeled M&Ms datasets, the results are as follows:
>
> | Methods | IBSSL (20% labeled Synapse) | SemiDG (2% labeled M&Ms) |
> | --| -- | -- |
> | A&D | 60.2 | 78.7 |
> | w/o SVDA | 55.3 | 76.5|
> | w/o Diffusion | 56.7 | 72.9|
> | w/o DRS | 52.0 | 76.8|
> | w/o RS | 58.7 | 77.1|
>
> On Synapse dataset for IBSSL, the results are slightly different with those in the UDA setting, the DRS plays more important role than the Diffusion. In 5% M&Ms dataset for SemiDG, the results are quite aligned with results in the UDA setting.

---

> > ### Comment · Area_Chair_9rnJ · 2023-08-18
> > **Rebuttal yd6u**
> >
> > Dear yd6u
> >
> > Could you have a look at the rebuttal to see if your questions have been clarified?
> >
> > Thanks,
> > Your AC

---

### Official Review · Reviewer_suEd · 2023-07-09

**Soundness:** 3 good
**Presentation:** 3 good
**Contribution:** 2 fair
**Rating:** 5
**Confidence:** 4

**Summary:**

The paper presents a novel framework for semi-supervised medical segmentation that seeks to address three different scenarios simultaneously: lack of labelled data (standard semi-supervised learning -- SSL), domain shifts (unsupervised domain adaptation -- UDA) and generalization to new, unseen domains (semi-supervised domain generalization -- SemiDG). Toward this goal, they combine several strategies in their framework: sampling-based data augmentation, diffusion V-Net for learning domain invariant representations, difficulty-aware supervised training to generate class-unbiased pseudo-labels, and a reparameterize and smooth (RS) technique based on on the Gumbel softmax & ensembling strategy to generate high-quality pseudo-labels. The proposed approach is tested on four different datasets and tasks: LASeg for the SS taskL, Synapse for class imbalanced SSL, MMWHS for UDA, and M&Ms for SemiDG. Results show the method to achieve a similar or better performance on all tasks compared to SOTA methods.




**Strengths:**

* The proposed framework brings together three different medical image segmentation scenarios involving unlabelled data: SSL, UDA and SemiDG. This bridges a gap in the literature, as existing approaches typically focus on a single of these scenarios.

* Authors do a good job in the Introduction (Figs 1-3) to situate their method with respect to existing approaches and highlight the limitations of these approaches.

* Experiments to evaluate the proposed method are comprehensive. The proposed method is evaluated in four different scenarios and, fr each one, compared against the SOTA. The experimental validation also includes several relevant ablation studies that demonstrate the usefulness of the method's different components.

* Results show the method to achieve SOTA performance on all tasks. These results are particularly impressive for the UDA task on the MMWHS (MR to CT) where the method achieves a performance on par with supervised training (more on this later).

**Weaknesses:**

* From my understanding, the technical novelty of the work comes from the combination of several existing techniques to address different problems in SSL. While some results are good, the proposed method feels heavily-engineered and complex.

* Experiments do not fully support the main claims of the paper. For example, current experiments employ different baselines to compare the method in the SSL, UDA and SemiDG settings, instead of testing the same approaches (or some of them) across these settings. Hence, the claim that current approaches can only handle one setting is not truly validated. Likewise, the claim that having separate branches for labelled and unlabelled is not well validated in the ablation study on architectures since all tested models implement this idea (as I understand)

*  In terms of results, the method yields more limited improvements for the SSL and SemiDG settings. It is unclear if these are statistically significant.

**Questions:**

* p2: "Currently, methods for these three scenarios are optimized separately, and there is no existing approach that addresses all three scenarios within a unified framework." As I understand, SSL and UDA are subcases of SemiDG, hence methods designed for the latter should be able to handle all three. I would relevant to compare the proposed approach against those in all three settings.

* p2: "Thus, the training process will easily get overwhelmed by the supervised training task." It is unclear to me why this is the case since unlabelled data is typically used to regularize the model trained with a supervised loss, and one of the main goals of regularization is to avoid overfitting. Authors should better validate this claim in their experiments.

* p5: " SVDA build upon an augmentation set and Naug operations are randomly sampled to apply to both the labelled and unlabelled data." This is in fact a very common strategy in SSL.

* Section 3.3: Normally, diffusion models require a lot of data to train. How is it possible to achieve this using only a few labelled samples ?

* Section 3.3: In the diffusion model, why do you add noise on the ground-truth segmentation maps? Assuming that the domain shift occurs in the image space (the anatomy should not vary from one modality or site to another), what can this achieve?

* Figure 4: I am confused about the diffusion encoder taking both a labelled and an unlabelled image as input, since they have different shapes. Can authors explain?

* Eq (2): the notation is confusing. Is 'e' the index of the sum or not?

* Eq (2): Technically, in the standard definition of PSI, you should have a Delta in the sum (as it is similar to a KL)

* Eq (2)-(3): As I understand, du_{k,e} is always negative while dl_{k,e} is always positive, hence w_diff should be negative?

* Eq (4): Why not use a standard focal loss to address this issue?

* p6: "we found that the denoised probability map pu;ψ is too sparse, i.e., with very high confidence of each class." Why not use a different softmax temperature or label smoothing to avoid this ?

* p7: For the class imbalanced scenario, why not used a class prior as in Bateson et al, since these priors are known from anatomical references

Bateson, Mathilde, Hoel Kervadec, Jose Dolz, Hervé Lombaert, and Ismail Ben Ayed. "Source-free domain adaptation for image segmentation." Medical Image Analysis 82 (2022): 102617.

* Table 3: the results for UDA on MMWHS are almost too good. It seems unlikely that the proposed method can achieve a Dice score similar to that of supervised trainining and a better ASD. Have authors checked for data leakage? Can they comment on this?













**Limitations:**

The discussion of limitations could be expanded (only the training time of the diffusion model is mentioned). Any failure cases?

---

> ### Author Rebuttal · Authors · 2023-08-10
>
> Sincere gratitude to reviewer suEd for the recognition of our idea. We hope our answers can address all your concerns. We are looking forward to having further discussions with you.
>
> **Q1: p2: "Currently, methods for …"**
>
> While current Semi-DG methods can be evaluated within SSL and UDA, they have yet to furnish results within these specific scenarios. To address this gap, we evaluate the state-of-the-art SemiDG methods in the SSL and UDA tasks, as shown below.
>
> For SSL on 10% labeled LASeg dataset:
> |Method| Dice|
> |--| --|
> |SS-Net (SOTA in SSL)|88.55|
> |vMFNet|73.88 |
> |EPL_SemiDG|76.49|
> |A\&D (ours)|89.40|
>
> For UDA of MR to CT on MMWHS dataset:
> |Method|Dice|
> |--|--|
> |LMISA-3D (SOTA in UDA)|81.5|
> |vMFNet|72.3|
> |EPL_SemiDG|71.9|
> |A\&D (ours)|90.0|
>
> This superiority of our approach over SemiDG methods in SSL and UDA settings can be attributed to the efficacy of our proposed SVDA, A&D strategy with diffusion, and DRS, which adeptly addresses scarcity of labeled data, domain shifts, and imbalanced class distributions.
>
> [1] Liu, et al. "vMFNet: Compositionality Meets Domain-Generalised Segmentation." MICCAI, 2022.
>
> [2] Yao et al.. "Enhancing pseudo label quality for semi-supervised domain-generalized medical image segmentation." AAAI, 2022.
>
> **Q2: p2: "Thus, the training process …**
>
> In theory, unlabeled data serves to mitigate overfitting. However, Figure 3 indicates that even when there is a significantly larger amount of unlabeled data compared to labeled data, the impact of unlabeled data on regularization is limited. This is due to the fact that labeled data, guided by precise masks for supervision, converges more rapidly.
>
> **Q3: p5: " SVDA build upon …**
>
> We agree that augmentation is a very common strategy. However, our SVDA, comprises a combination of 3D transformations, is designed to empower Diffusion-vnet in generating a more diverse dataset. It enhances feature diversity within the limited medical images and aids in fully harnessing the capabilities of our diffusion-based A&D framework across various scenarios.
>
> **Q4: Normally, diffusion models …**
>
> While it's true that generative diffusion models often demand a substantial amount of data for training, the scenario changes when employing them for perceptual tasks in volumetric medical image segmentation. In such cases, even a limited number of samples can yield favorable performance [3], which is due to the inherent nature of volumetric samples, wherein a single sample typically includes over 200 2D slices.
>
> [3] Xing et al. "Diff-UNet: A Diffusion Embedded Network for Volumetric Segmentation." arXiv, 2023.
>
> **Q5: In the diffusion model …**
>
> Adding noise to the label can perform better in perceptual tasks [3,4]. Following [3, 4], we concatenate the noisy label with image embeddings, the diffusion model needs to denoise these concatenated noisy features and recover the clear label. Thus, the diffusion model can jointly learn the image-level feature.
>
> [4] Ji et al. DDP: diffusion model for dense visual prediction. ICCV, 2023
>
> **Q6: Figure 4: I am confused …**
>
> The training processes of the labeled and unlabeled data are sequential. In each training iteration, the model first takes the labeled data to train the modules highlighted in blue within Figure 4, then generates the pseudo labels. Finally, the model takes the unlabeled data to train the encoder again with the orange-colored decoder. We employ if-else statements and different embedding layers to handle variations in shapes.
>
> **Q7: Eq (2): the notation …**
>
> Yes, 'e' is the index of the sum to indicate sum from $e-\tau$ iteration to e iteration.
>
> **Q8: Eq (2): Technically …**
>
> Sorry, it is a mistake, $\mathbb{I}(\cdot)$ function should be $\mathbb{I}_u(x)=min(x,0)$ and  $\mathbb{I}_l(x)= max(x,0)$ .
>
> **Q9: Eq (2)-(3): As I understand …**
>
> Actually, $du_{k,e}$ and $dl_{k,e}$ are both always positive since in $du_{k,e}$, $\Delta<0$, $ln(\cdot)<0$, so $du_{k,e}>0$; in $dl_{k,e}$, when $\Delta>0$, $ln(\cdot)>0$, so $dl_{k,e}>0$. Hence, $w_{diff}$ is always positive.
>
> **Q10: Eq (4): Why not use …**
>
> The focal loss relies on a predefined class distribution. However, in SSL tasks, only the class distribution of labeled data is known and constant, while the distribution of unlabeled data remains uncertain and evolves dynamically throughout training. Consequently, employing a standard focal loss directly is unsuitable for addressing imbalanced SSL tasks.
>
> **Q11: p6: "we found that …**
>
> We thank the reviewer for the suggestion. We add new ablation experiments by replacing the our RS with softmax temperature and label smoothing. The results are shown below:
>
> | | Dice|
> |--|--|
> |A\&D w/ RS (ours)|90.0|
> |A\&D w/o $p^{u;\xi}$(lower bound)|87.3|
> |A\&D w/ softmax temperature 2|88.0|
> |A\&D w/ softmax temperature 4|87.7|
> |A\&D w/ label smoothing|86.5|
>
> The degradations of the performance are due to the fact that these smoothing strategies lead to a reduction in the confidence of true positive regions. As a result, when combined with $p^{u;\psi}$ to obtain the pseudo label, $p^{u;\xi}$ will be overwhelmed.
>
>
> **Q12: p7: For the class imbalanced scenario ...**
>
> The class prior approach requires a substantial amount of prior knowledge, demanding task-specific ratio computations. While it might lead to improved performance, it lacks generality, as obtaining the class prior for certain tasks could be challenging. In contrast, our A&D framework yields favorable results in a wider range of practical scenarios.
>
> **Q13: Table 3: the results …**
>
> We have reviewed the code carefully and are confident that no data leakage issues are present. Upon acceptance, we will release both our code to the public.
>
> **Limitations**
>
> One limitation lies in the modest enhancements over the SemiDG benchmarks. This limitation arises from the MRI data in the SemiDG dataset, which possesses restricted depth dimension (less than 10). Consequently, the performance of our model, founded upon the 3D V-net architecture, is limited.

---

> > ### Comment · Reviewer_suEd · 2023-08-11
> > **Thanks for the detailed answers**
> >
> > I have read the authors' answers to my comments and, overall, am satisfied with them. I still have some remaining concerns with training a diffusion model with very little data. Can the authors comment on how much data is required and what happens if not enough data is provided? Also, it would be interesting to discuss failure cases of the method.

---

> > > ### Author Response · Authors · 2023-08-11
> > >
> > > We thank reviewer suEd for the quick reply and the recognition of our comments.
> > >
> > > Regarding training a perceptual diffusion model with very little data, a noteworthy distinction emerges when compared with their generative counterparts. Traditional generative diffusion models recover clear **images** from noisy images. In contrast, the perceptual diffusion models recover clear **labels** from a combination of the noisy labels and image embeddings, which is a much easier task since the model only focuses on recovering the shapes rather than the detailed textures. Thus, the model can be trained with very few **well-labeled** data in volumetric medical image segmentation since the anatomical structures are consistent across samples.
> > > The most extreme case in this work is the 2% labeled SemiDG task, with only **eight labeled data samples** (about 80 2D slices). The diffusion model in our framework works well in this scenario. Based on our experience, label **quality** is more important than data **quantity** to train diffusion models in medical images. However, using insufficient data to train the diffusion model in natural images, characterized by intricate scenarios and varying segmentation targets per sample, is prone to result in non-convergence issues.
> > >
> > > Regarding the failure cases, as we mentioned in the Limitations of the comments, they are mainly in the M&Ms dataset for SemiDG setting. Specifically, our method usually fails on the **first and the last slices along the depth axis**. Due to the restricted depth dimension (less than 10), the 2D slices and the corresponding masks vary significantly (analogy deleting frames from a video sequence). In such a case, it is hard for our volumetric framework to capture depth-wise information for the first or last slice with only one neighboring slice as a reference, and thus leads to false positive results.

---

### Decision · Program_Chairs · 2023-09-21

**Decision:**

Accept (poster)

**Comment:**

The paper introduces a semi-supervised learning (SSL) for 3D medical image segmentation to work in three scenarios: 1) SSL and imbalanced SSL; 2) unsupervised domain adaptation (UDA); and 3) semi-supervised domain generalisation. The proposed method combines data augmentation, diffusion model with V-Net architecture for domain invariant representations, difficulty-aware supervised training to produce class-balanced pseudo-labels, and an approach to generate high-quality pseudo-labels. Four datasets have been used in the experiments, namely: LASeg for the SS taskL, Synapse for class imbalanced SSL, MMWHS for UDA, and M&Ms for SemiDG. Results on these datasets demonstrate that the proposed approach has competitive or better performance than the SOTA methods. The paper received the following scores: 7,5,5,4,4. The main strengths of the paper are: 1) interesting unification of the SSL, UDA and SemiDG tasks; 2) nice motivation for the paper; 3) comprehensive evaluation; 4) SOTA performance on all tasks; and 5) well-written. The main weaknesses found by reviewers were: 1) method looks heavily-engineered and complex; 2) unclear description of the method; 3) unclear translation to tasks outside medical image analysis; 4) unclear if claims are too broad; 5) experiments missing SOTA results on the LA dataset; 6) unclear take-home message to explain the SSL SOTA results; 7) several related works are missing [1,2,3,4,5,6,7,8]. The authors did a good job to clarify the issues in the rebuttal. In general, the paper shows a solid approach, with good results. However, the technique seems to be incremental without a clear contribution.

[1] Gu et al. “ConFUDA: Contrastive Fewshot Unsupervised Domain Adaptation for Medical Image Segmentation”, 2022 [2] Xia et al. “Uncertainty-aware multi-view co-training for semi-supervised medical image segmentation and domain adaptation”, 2020 [3] Bian et al. “DDA-Net: Unsupervised cross-modality medical image segmentation via dual domain adaptation”, 2021 [4] Perone et al. “Unsupervised domain adaptation for medical imaging segmentation with self-ensembling”, 2019 [5] Shin et al. “COSMOS: Cross-Modality Unsupervised Domain Adaptation for 3D Medical Image Segmentation based on Target-aware Domain Translation and Iterative Self-Training”, 2022 [6] Bateson et al. “Source-Relaxed Domain Adaptation for Image Segmentation”, 2021 [7] Liu et al. “S-CUDA: Self-cleansing unsupervised domain adaptation for medical image segmentation”, 2021 [8] Liu et al. “ACT: Semi-supervised Domain-adaptive Medical Image Segmentation with Asymmetric Co-Training”, 2022